# CityGPT: Generative Transformer for City Layout of Arbitrary Building Shape

## Abstract

City layout generation has gained substantial attention in the research community with applications in urban planning and gaming. We introduce CityGPT, the generative pre-trained transformers for modeling city layout distributions from large-scale layout datasets without requiring priors like satellite images, road networks, or layout graphs. Inspired by masked autoencoders (MAE), our key idea is to decompose this model into two conditional ones: first a distribution of buildings' center positions conditioned on unmasked layouts, and then a distribution of masked layouts conditioned on their sampled center positions and unmasked layouts. These two conditional models are learned sequentially as two transformer-based masked autoencoders. Moreover, by adding an autoregressive polygon model after the second autoencoder, CityGPT can generate city layouts with arbitrary building footprint shapes instead of boxes or predefined shape sets. CityGPT exhibits strong performance gains over baseline methods and supports a diverse range of generation tasks, including 2.5D city generation, city completion, infinite city generation, and conditional layout generation.

## 1 Introduction

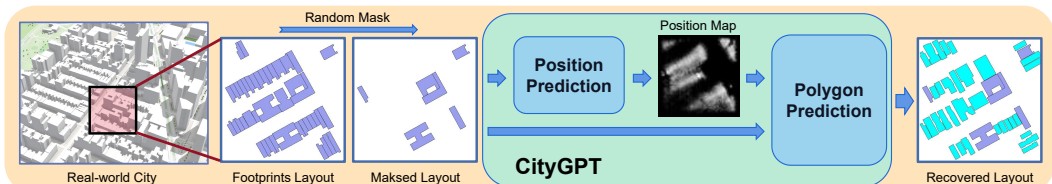

Figure 1: **Overview of CityGPT** that generates 2D footprints as city layouts. The left image displays a real-world city scene, with our model addressing the footprint layout of specific city blocks. Our model is trained to reconstruct the original layout based on a few unmasked buildings.

City layout generation is a vital research area dedicated to automating the creation of realistic and coherent building arrangements. While generating the entire 3D city structure presents challenges, many studies focus on generating city footprints in vectorized form. This vectorized representation, as opposed to rasterized data, offers greater accuracy and conciseness, enabling the depiction of diverse symbols such as drawings (Ha & Eck, 2017), 3D wireframe structures (Nash et al., 2020), and more. Due to the high dimensionality of vectorized data, this task remains challenging.

Prior works have approached city layout modeling in various ways. Some methods incorporate computer vision recognition into the modeling process by reconstructing building layouts from satellite, aerial imagery, or road networks (Cheng et al., 2019; Demir et al., 2018; He et al., 2023b; Li et al., 2021; Mahmud et al., 2020; Musialski et al., 2013; Xu et al., 2022b; Zorzi et al., 2022; Lin et al., 2023; He & Aliaga, 2023). Urban procedural modeling is another approach that generates city components with expert knowledge (Müller et al., 2006; Chen et al., 2008; Nishida et al., 2016; Vanegas et al., 2012; Benes et al., 2021; Aliaga et al., 2008; Lipp et al., 2011; Merrell et al., 2010; Feng et al., 2016). There are also many works (Xu et al., 2021; He & Aliaga, 2023) that focus solely on city layout generation. While these methods can generate realistic city layouts, they heavily rely on complex priors, making them unsuitable for scenarios where such prior information is not available.

In our work, we introduce CityGPT, a transformer-based model to advance city layout generation capabilities. Our model draws inspiration from three key observations:

**1.** Transformers (Vaswani et al., 2023) are renowned for their ability to capture relations between elements, which aligns well with city layouts since buildings are strongly interconnected. Transformers are also capable of handling varied length data, suitable for the variable number of buildings and edges in building footprints. Additionally, the scalability of transformers during inference can allow us to simplify the training procedure design. **2.** In many city generation scenarios, such as completing a city, extending a city, or generating a city from scratch, they all share a common essential requirement: the need to generate buildings from a given set of buildings. Inspired by this, we have designed a masked training strategy that has been explored in the field of natural language processing (NLP) (Devlin et al., 2019) and computer vision (He et al., 2021). This pipeline design allows for various user-generated choices within a single training mode. **3.** The human construction process for real-world cities involves selecting site positions and then designing buildings. Following this natural human design procedure, our model consists of two stages: first generating positions and then generating building footprints.

Our model mainly consists of two stages, as shown in Figure 2. The first stage conditions on given existing buildings and generates positions for additional buildings, while the second stage conditions on the existing buildings and predicted positions to generate building footprints. After training, our iterative generation process can produce realistic arbitrary city layouts, consisting of any number of buildings with arbitrary shapes. Moreover, due to the structure design of the model and the iterative generation process, users can easily achieve non-overlapping generation and control the generation procedure at any time by regenerating certain buildings while maintaining the past generated buildings, adding building footprints to the entire city layout, or masking undesired areas.

To the best of our knowledge, our method is the first to generate city layouts of arbitrary scales, consisting of buildings with arbitrary shapes, without any prior conditions. Additionally, our model can learn a strong representation of city layouts, as demonstrated by its performance in the classification task. We adapt several prior related methods (Ha & Eck, 2017; Jyothi et al., 2021; Gupta et al., 2021; Han et al., 2023; Inoue et al., 2023) to our specific task setting and compare our method qualitatively and quantitatively with them. We show that none of them support arbitrary building shapes and infinite city layout generation from scratch. Furthermore, we showcase our model's capacity in various tasks, such as 2.5D city generation, infinite city generation, and city complementation. Our primary contributions encompass the following:

**1.** We introduce a two-stage decomposition modeling approach for city layout and design a transformer-based model with a masked training strategy. These innovations enable our model to accomplish various city layout generation tasks. These tasks include, but are not limited to, from-scratch generation, generating infinite variations, human-guided generation, city completion, and more. **2.** Our model also provides an effective way to learn a good city layout representation. In the downstream classification task, our pre-trained model can enhance both the performance and convergence rate of the vanilla benchmark. **3.** We provide adaptations of various existing methodologies in city layout generation, with the overarching goal of enhancing and propelling future research endeavors in this research area.

## 2 RELATED WORKS

**City Generation.** City modeling is a multifaceted field that has witnessed diverse approaches. One common approach integrates computer vision techniques into the modeling process, reconstructing building layouts from satellite or aerial imagery (Cheng et al., 2019; Demir et al., 2018; He et al., 2023b; Li et al., 2021; Mahmud et al., 2020; Musialski et al., 2013; Xu et al., 2022b; Zorzi et al., 2022; Lin et al., 2023). Their innovative methodologies use image recognition to convert visual data into city layouts. Urban procedural modeling (Müller et al., 2006; Chen et al., 2008; Nishida et al., 2016; Vanegas et al., 2012; Benes et al., 2021; Aliaga et al., 2008; Lipp et al., 2011; Merrell et al., 2010; Feng et al., 2016), on the other hand, focuses on generating city components through expert knowledge. It employs rule-based or algorithmic frameworks to simulate urban element creation, like buildings, roads, parks, and architectural features. While this expert-driven strategy has been influential in producing realistic city models, it is heavily reliant on expert knowledge, limiting its application in scenarios lacking comprehensive prior information. The central aim of our research

is to democratize city design. We aspire to develop an approach that empowers novices, enabling them to create captivating city layouts without being encumbered by intricate prerequisites.

**Layout Generation.** Beyond the realm of city modeling, a plethora of works have concentrated on layout generation across diverse domains. These include document layout (Patil et al., 2020), graph layout (Lee et al., 2020), and in-

door scene layout (Ritchie et al., 2019; Wang et al., 2018; 2019), employing a spectrum of techniques such as Variational Autoencoder (VAE) (Jyothi et al., 2021; Arroyo et al., 2021; Han et al., 2023), Generative Adversarial Network (GAN) (Li et al., 2019; Kikuchi et al., 2021), Transformer (Gupta et al., 2021; Arroyo et al., 2021; Yang et al., 2021; Kong et al., 2022), and diffusion (Inoue et al., 2023; He et al., 2023a; Zhang et al., 2023). A subset of these methodologies treats layout as a graph structure (Nauata et al., 2020; Xu et al., 2021; Bao et al., 2013; He & Aliaga, 2023; Para et al., 2020; Chang et al., 2021), and some works (Zheng et al., 2019; Jiang

|  | Arb. Num. | Arb. Shape | Order-free | Infinite | Uncond. |
|---|---|---|---|---|---|
| SketchRNN (Ha & Eck, 2017) | ✔ | ✔ | ✗ | ✗ | ✔ |
| LayoutVAE (Jyothi et al., 2021) | ✔ | ✗ | ✔ | ✗ | ✔ |
| LayoutGAN++ (Kikuchi et al., 2021) | ✗ | ✗ | ✔ | ✗ | ✔ |
| LayoutTrans (Gupta et al., 2021) | ✔ | ✗ | ✗ | ✗ | ✔ |
| AETree (Han et al., 2023) | ✗ | ✗ | ✔ | ✗ | ✔ |
| LayoutDM (Inoue et al., 2023) | ✔ | ✗ | ✔ | ✔ | ✔ |
| LayoutTrans++ (Jiang et al., 2023) | ✔ | ✗ | ✗ | ✔ | ✔ |
| GlobalMapper (He & Aliaga, 2023) | ✔ | ✗ | ✔ | ✔ | ✗ |
| Ours | ✔ | ✔ | ✔ | ✔ | ✔ |

Table 1: **Comparison with Existing Methods**: Each column represents a generation capacity: arbitrary building numbers, arbitrary building shapes, unordered layout generation, infinite generation, and unconditional generation. Our method effectively addresses all of these generation options, surpassing the capabilities of existing methods.

et al., 2023; Chai et al., 2023) focus on generating layouts based on content or human constraints. However, these existing methods often grapple with inherent limitations pertaining to generation scale, shape, or predefined graph connection conditions. These constraints can hinder their applicability in various creative scenarios. In comparison to established techniques, our approach showcases more diversified generation choices, as evidenced by the comparison presented in Table 1.

**Transformer Based Model.** Transformer has emerged as a prominent and influential approach across various research areas. Transformer-based models, such as GPT (Radford et al., 2018), have made significant strides in NLP tasks, with subsequent works (Radford et al., 2019; Brown et al., 2020) building upon its success and achieving highly effective results. In the field of computer vision, the Masked Autoencoder (MAE) (He et al., 2021) also based on the transformer architecture has demonstrated substantial progress. Its masked learning structures have been demonstrated to be remarkably effective across a diverse range of tasks, including, but not limited to (Guo et al., 2023; Pang et al., 2022; Wang et al., 2023). Additionally, it has shown prowess in generation tasks (Li et al., 2023). The remarkable versatility of the MAE's architecture makes it a powerful and adaptable tool applicable across numerous domains. In our work, we designed a transformer-based model and a masked training strategy inspired by the above works. Leveraging these, our model has shown remarkable effectiveness in diverse city layout generation scenarios and downstream tasks.

## 3 METHOD

### 3.1 PROBLEM FORMULATION

In our task, each building is represented as a 2D polygon, and we focus on generating building sets within certain city blocks. A building set, denoted as $B = \{f_k\}_{k=1}^N$, and $f_k = [(x_k^1, y_k^1), (x_k^2, y_k^2), \ldots, (x_k^{n_k}, y_k^{n_k})]$, where each $f_k$ represents a certain building's footprint, and $N$ is the number of buildings inside the block. Each block is a $500 \times 500$ m$^2$ square in the real world. For each footprint $f_k$, $x_k^i, y_k^i \in \mathbb{R}$, where $0 < x_k^i, y_k^i < 500$, denotes the relative position of the $i$-th vertex with respect to the block's origin, and $n_k$ represents the number of edges in footprint $f_k$. Additionally, we introduce a position set $P = \{p_k\}_{k=1}^N$, where $p_k = \frac{1}{N} \sum_{i=1}^{n_k} (x_k^i, y_k^i)$ represents each building's position by its mean center. Under this task setting, *our method's key idea is to model the distribution of building sets as the following conditional distribution decomposition*:

$$\mathcal{P}(B) = \mathcal{P}(\bar{B}_i \mid B_i)\mathcal{P}(B_i) = \mathcal{P}(\bar{B}_i \mid B_i, \bar{P}_i)\mathcal{P}(\bar{P}_i \mid B_i)\mathcal{P}(B_i \mid \emptyset, P_i)\mathcal{P}(P_i \mid \emptyset) \quad (1)$$

Here, $B$ represents the entire building set, and $B_i$ represents any non-empty subset of $B$. $\bar{B}_i$ represents the complementary set of $B_i$. $P_i$ and $\bar{P}_i$ are the position sets corresponding to $B_i$ and $\bar{B}_i$,

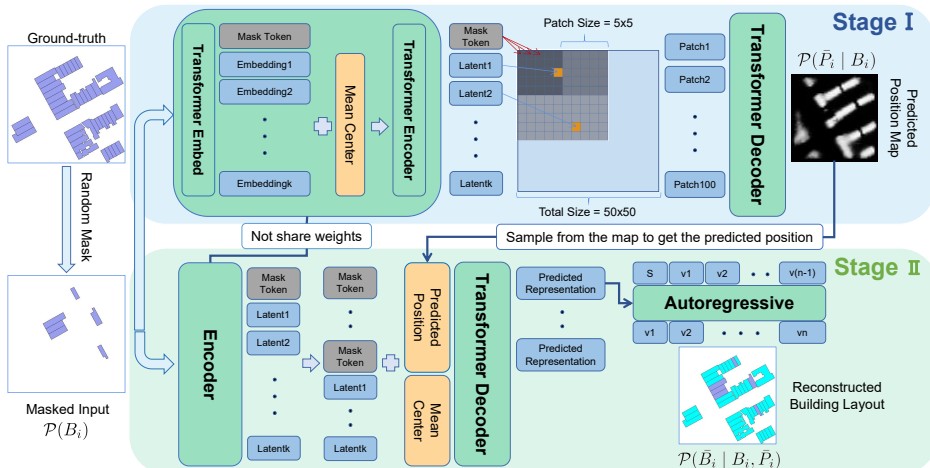

Figure 2: **CityGPT Pipeline.** Our approach to city layout generation involves a two-stage process. In Stage I, we focus on reconstructing the positional probability map for the target buildings. In Stage II, we then proceed to reconstruct the actual buildings based on the provided positions.

respectively. Thus, our task can be divided into two main parts. The first part involves generating the complementary position set given a polygon subset:

$$\mathcal{P}(\bar{P}_i \mid B_i) \tag{2}$$

The second part involves generating the complementary polygon set given a polygon subset and the complementary position set:

$$\mathcal{P}(\bar{B}_i \mid B_i, \bar{P}_i) \tag{3}$$

By successfully modeling these processes, we can decompose the objective into a superposition of sub-processes, enabling us to generate the entire set iteratively, one by one or set by set.

### 3.2 STAGE I: POSITION PREDICTION

Drawing inspiration from the design of MAE (He et al., 2021), we have devised a transformer-based encoder and decoder structure to address the position prediction objective (Eq. 2), as depicted in Figure 2. Here's a breakdown of our model structure:

**1. Random Masking:** Initially, we randomly mask some polygons from a block. The remaining polygons are then input into the transformer encoder, which produces a latent representation of these buildings. **2. Discretization:** The entire block is discretized into a $50 \times 50$ mesh, specifically used for position prediction. In our dataset, a block represents roughly a 500-meter area in the real world, and over $99.9\%$ of buildings satisfy the $50 \times 50$ discretized condition. This ensures that almost no two buildings appear in the same grid, which need to be discarded. **3. Latent Representation to Grid:** The encoder's latent representations are placed into the corresponding grid positions within the $50 \times 50$ mesh. **4. Patchify:** To reduce the sequence length for the decoder transformer, we employ a patchifying process on the $50 \times 50$ mesh before feeding it into the decoder. The decoder then produces the output, which is unpatchified to obtain the final position prediction.

### 3.3 STAGE II: POLYGON PREDICTION

Similar to adapting the MAE (He et al., 2021) structure for the position prediction task, we also employ a transformer-based encoder and decoder for polygon prediction given positions (Eq. 3). The model structure is depicted in Figure 2. Here is a breakdown of our approach:

**1. Random Masking:** As in Stage I, we start by randomly masking some buildings. **2. Adding Position Embedding:** The transformer decoder takes the desired position sets $\bar{P}_i$ as input, which are continuous 2D coordinates. To incorporate this positional information, we utilize the position embedding method described in (Vaswani et al., 2023). This embedding process maps both the encoder position sets $P_i$ and the complementary position sets $\bar{P}_i$ into the same dimension as the

decoder latent space. Then add them to the corresponding encoder latent variables or mask tokens.
**3. Autoregressively Generating Polygons:** Once we have the decoder's output representing each polygon, we concatenate this representation with all inputs of the autoregressive model to create the polygons. And the condition Stage II contains sufficient information, enabling us to consider the probability (Eq. 3) as a function $\delta(\hat{B})$ at a certain polygon set $\hat{B}$. This means the autoregressive model can directly generate all continuous coordinates of the polygons, rather than probabilities over discretized coordinates. This approach, as demonstrated later, can generate an infinite variety of results. If necessary, our model can also readily adapt to the standard sample generation pattern by discretizing polygon vertices, as discussed in Appendix G.

### 3.4 Training & Inference

**Training:** Our model adopts two training pipelines. In the first stage, our model predicts 2500-dimensional values between $[0, l]$, representing the probabilities for each position to have a building. We use the Binary Cross Entropy (BCE) loss between the predictions and the target position sets $\bar{P}_i$. To address the class imbalance issue where the number of samples with label 0 and 1 may differ significantly, we incorporate a positive weight in the BCE loss to give more importance to the minority class. During the second stage training, we use ground truth positions in the decoder process. For each polygon, we employ the translation of the polygon vertices sequence to train the autoregressive model. Its output contains three dimensions: two for coordinates and one for the end token. The loss function comprises two terms. The first is L2 Loss (Mean Squared Error), which measures the mean squared difference between the predicted coordinates and the ground truth coordinates. The second is BCE Loss (Binary Cross Entropy), which is applied to the end token output of the autoregressive model.

**Inference:** In our generation process, we generate the buildings one by one. Initially, we uniformly sample a position from the range $[0, 50] \times [0, 50]$. Subsequently, we iteratively utilize the first and second stage models to generate all the buildings. Throughout this generation process, we implement several validation steps. Firstly, in most cases (with a probability of 0.9), we determine the next position by selecting the one with the highest probability. Since the initial position is sampled from a continuous space, each generated result is unique, and we can also demonstrate that it maintains generation diversity by considering later metrics. Secondly, we discard positions that would generate polygons overlapping with previous ones (if not, the overlap ratio is about 7%). Thirdly, we end our generation process if all valid positions have predicted probabilities less than 0.5.

## 4 Experiments

### 4.1 Experiments Detail

**Datasets.** We collected datasets from Manhattan, covering 40,000 city blocks and 45,847 buildings. The data was split into three parts, with 80%/10%/10% allocated for training, validation, and testing, respectively. In the extraction process, we obtained blocks by selecting a fixed-size block centered on each building, with a block size of 500 meters. Blocks with fewer than 6 buildings were excluded from the dataset. Additionally, we simplified all polygons by reducing vertices to less than 20. This was achieved by removing nodes with minimal triangle areas (the area of the triangle formed by the node and its neighbors). This simplification had minimal observed impact on the original data.

**Training Detail.** We executed two end-to-end training pipelines corresponding to our two stages, respectively. Both pipelines were trained on a single RTX8000 GPU, utilizing a learning rate of 0.001. Each stage required approximately 1 day to converge.

### 4.2 Generation Results & Comparison

We provide quantitative and qualitative comparison between our model and some relative work. The specific implementation detail of baselines can be found in Appendix E.

**Quantitative Results.** For quantitative comparisons with other models, we use six metrics to highlight the effectiveness of our approach. CitySim: Inspired by DocSim (Patil et al., 2020), we designed CitySim to evaluate the similarity between pairs of city layouts. The specific formula for

| | CitySim ↑ | IPR ↓ (%) | FID ↓ | WD ↓ (edge) | WD ↓ (area) | WD ↓ (ratio) |
|---|---|---|---|---|---|---|
| SketchRNN (Ha & Eck, 2017) | 0.070 | 3.88 | 149.1 | 0.856 | 429.5 | 0.0983 |
| LayoutVAE (Jyothi et al., 2021) | 0.013 | 0.255* | 222.6 | 2.64 | 1018 | 0.247 |
| LayoutTrans (Gupta et al., 2021) | 0.085 | 1.45 | **48.44**† | 1.04 | 436.1 | 0.0786 |
| AETree (Han et al., 2023) | 0.041 | **0.218*** | 153.3 | 3.16 | 1674 | 0.309 |
| LayoutDM (Inoue et al., 2023) | 0.068 | 9.38 | 100.5 | 1.26 | 319.2 | 0.0881 |
| **Ours** | **0.098** | 0.471 | 53.11 | **0.426** | **130.3** | **0.0777** |

Table 2: **Quantitative Comparison.** We generated 1000 city layouts and compared them to the same number of real city layouts. The best values are in bold, and the second-best values are underlined. ∗: LayoutVAE and AETree excel in IPR because they still produce simple polygons, limiting their complexity, as seen in the third metric. †: Although LayoutTrans achieves the best FID score, its generation is constrained by order-dependent attributes, as explained further in Section 4.2.

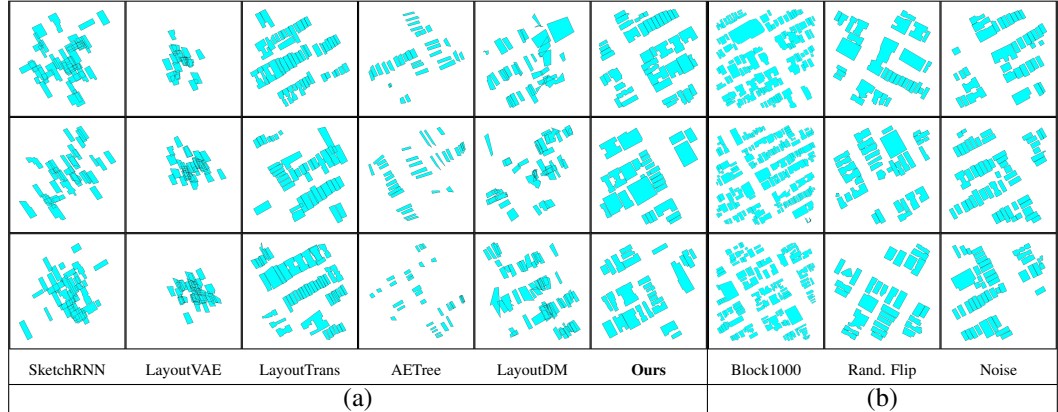

Figure 3: **(a) Qualitative Comparison.** It is evident that both SketchRNN and LayoutVAE struggle to capture the intricate relationships between buildings. AETree struggles to generate diversified polygons. Although LayoutTransformer manages to produce visually appealing outcomes, it remains reliant on predefined order constraints, which hampers its adaptability to diverse generation scenarios. LayoutDM encounters difficulties in producing valid polygons. **(b) Qualitative Results of Robustness Study.** From left to right is the results of extending the block size, adding random flip, and adding random noise respectively.

CitySim can be found in Appendix C. IPR (Invalid Polygon Ratio): IPR represents the percentage of generated polygons that are invalid due to self-intersections, having less than three nodes, etc. FID (Fréchet Inception Distance) (Heusel et al., 2017): FID is applied to rendered images at a pixel resolution of $500 \times 500$, with buildings represented in blue and edges in black. WD (Wasserstein Distance): Calculated in three scenarios: number of edges of each building, area of each building, and ratio of the area occupied by buildings to the total block area. All these five metrics are computed across 1000 generated results. The comparison results are presented in Table 2.

**Qualitative Results.** The qualitative comparison is illustrated in Figure 3 (a). It is evident that only the LayoutTransformer can produce results that are comparable to our model in the context of city layout generation. However, the LayoutTransformer heavily relies on manually specified prior settings for building order, a limitation commonly observed in autoregressive models. As a result, it cannot exhibit the diverse generation capabilities within a single training phase that we will demonstrate in Section 5, especially regarding infinite city generation and city complementation.

**User Study.** We conducted a user study to evaluate the perceptual realism of our method. Each user was presented with 20 single-choice questions, each offering three options: the ground-truth layout with minor noise (as the ground-truth data is often perfectly aligned, making it easy to distinguish; the noise scale is shown in Figure H.1), the layouts generated by our model, and those by LayoutTrans (Gupta et al., 2021) (the only baseline method achieving comparable visual results). Users had to select the most realistic option among these three choices. We received 24 submissions from different users, totaling 480 responses. The results indicated that 36.68%, 47.49%, and 15.83%

|     |                            | BCE ↓       | FID ↓       | WD ↓ (edge) | WD ↓ (area) | WD ↓ (ratio) |
| --- | -------------------------- | ----------- | ----------- | ----------- | ----------- | ------------ |
| (a) | Ours−Decoder               | 0.82        | 109.3       | 0.450 | 326.6      | 0.0911 |
|     | Ours−Patchify              | 0.75 | 62.21 | 0.533      | 219.6 | 0.0950     |
|     | Ours                       | **0.60**    | **53.11**   | **0.426**   | **130.3**   | **0.0777**   |
|     |                            | IPR ↓ (%)   | FID ↓       | WD ↓ (edge) | WD ↓ (area) | WD ↓ (ratio) |
|     | Combine Two Stages         | 0.494       | 71.34       | 0.512       | 286.9       | 0.101        |
|     | Ours − VMT                 | 0.612       | 54.91       | 0.465 | 166.6      | 0.0831       |
| (b) | Ours ($RN_1$=$RN_2$=6-15)  | 0.585       | 64.35       | 0.787       | 323.5       | 0.188        |
|     | Ours ($RN_1$=0-6,$RN_2$=6) | **0.363**   | 48.63 | 0.529      | 231.0       | 0.0901       |
|     | Ours ($RN_2$=6,$RN_2$=0-6) | 0.501       | 54.01       | 0.506       | 164.4 | **0.0770** |
|     | Ours ($RN_2$=$RN_2$=0-6)   | 0.421 | **44.99**  | 0.544       | 260.2       | 0.0976       |
|     | Ours ($RN_2$=$RN_2$=6)     | 0.471       | 53.11       | **0.426**   | **130.3**   | 0.0777 |

Table 3: **(a) Ablation Study for Stage I.** The initial BCE metric is the Stage I loss function described in Section 3.4, with a positive weight of 100. **(b) Ablation Study.** The best values are in bold, and second best values are underlined. For detailed experiment explanations, refer to Section 4.3.

of the responses considered the ground-truth (with minor noise), Ours, and LayoutTrans to be the most realistic, respectively. We also conducted a user study without noise on the ground-truth (21 submissions), and the results were 51.65%, 31.67%, and 16.68%, respectively. Regardless of the setting chosen, the results show that our generation results are most similar to the ground-truth.

## 4.3 ABLATION STUDY

We have conducted a series of ablation studies to showcase the sensitivity of key components in our model. Additional ablation studies regarding our overall model architecture design can be found in Appendix F. To begin, let's focus on the first stage of our model. We compared our approach against two variants: one without our masked decoder, utilizing a direct MLP to decode the transformer encoder's output; and another without the patchify setting, where the decoder directly accepts inputs with a sequence length of $50 \times 50$. Both quantitative outcomes (Table 3 (a)) and qualitative comparisons (Figure 4) underscore the effectiveness of our final model structure.

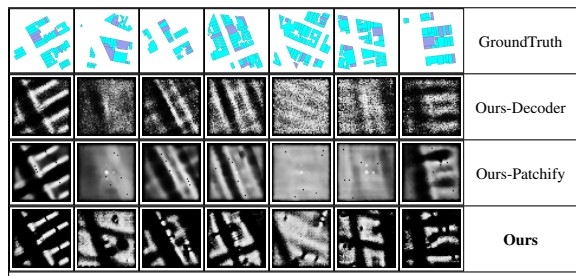

Figure 4: **Qualitative Comparison for Stage I.** The first row displays the ground truth city layout, with purple indicating existing buildings and blue representing those that need to be predicted. In the subsequent rows, whiter areas indicate a higher probability of buildings.

We also conducted comparisons involving several crucial structural and parameter settings within our complete model framework. These comparisons include:

**1. Combined Two Stages:** This involved merging our two stages into a single stage, sharing the transformer encoder and decoder, while segregating the output MLP into two parts corresponding to positional and polygon outputs respectively. **2. Ours - VMT:** In this variation, instead of using the first encoder output as the mask token, the model without VMT (Variant Masked Token) directly employed a learnable parameter as the masked token, akin to the original MAE approach (He et al., 2021). **3. Varying Remaining Building Numbers:** We present results for various remaining building numbers during the training process. $RN_1$ and $RN_2$ denote the remaining building numbers for stages I and II, respectively. The range 6-15 indicates that the remaining number varies from 6 to 15 during training, as does the range 0-6.

The outcomes, as presented in Table 3 (b), demonstrate the effectiveness of our model structure when compared to scenarios 1 and 2. Additionally, in terms of the remaining building number parameter, our experiments reveal that a large remaining number does not yield satisfactory results. And employing a small remaining number can enhance overall performance metrics like the FID score. However, it can compromise the diversity of generated polygons, as evidenced by the higher values in the WD metric for polygon edges and areas.

## 4.4 STUDY OF ROBUSTNESS

We conducted additional experiments to elucidate the robustness of our model, as outlined below:

**1. Extending the Block Size:** Expanding the block size benefits our model by capturing more global information. Importantly, our model structure supports larger block sizes without substantial computational costs, thanks to the patchify operation in the first stage, enabling seamless scalability. In the second stage, the sequence length aligns with the number of buildings, remaining within acceptable limits even with larger block sizes. This contrasts with other models like LayoutTransformer (Gupta et al., 2021) or Lay-

|  | IPR ↓ (%) | FID ↓ | BCE ↓ | Recon. L1 ↓ | Recon. Len. ↓ |
|---|---|---|---|---|---|
| Ours + Block1000 | 1.89 | 65.75 | 0.35 | 10.2 | $1.4 \times 10^{-3}$ |
| Ours + Rand. Flip | 0.783 | 55.13 | 0.64 | 12.3 | $1.3 \times 10^{-3}$ |
| Ours + Noise | 0.347 | 62.24 | 0.59 | 7.1 | $1.0 \times 10^{-3}$ |
| Ours | 0.471 | 53.11 | 0.60 | 4.5 | $3.5 \times 10^{-4}$ |

Table 4: **Quantitative Results of Robustness Study.** BCE , Recon. L1, and Recon. Len. are the loss value described in Section 3.4. For detailed experiment explanations, refer to Section 4.4.

outDM (Inoue et al., 2023), which rely on discretized coordinates and flattened sequence lengths. They face significantly higher computation costs when extending their block size. For instance, with a block size of 1000 meters, the sequence length can exceed 7,000. **2. Adding Random Flip:** We implemented a data preprocessing step that involves randomly flipping the blocks along their x or y directions. **3. Adding Random Noise:** Another preprocessing step we employed involved the introduction of random noise to the polygons.

Quantitative and qualitative outcomes from these experiments are presented in Table 4 and Figure 3 (b). These results collectively exhibit a reasonable and acceptable performance.

## 5 FURTHER EXPERIMENTS: MORE CHALLENGING SETTINGS

**Infinite City Generation.** Our model effortlessly generates consistent and infinite city layouts using the same training pipeline and results as previous. By smoothly shifting the generated window, we can continuously produce endless city layouts, as visually demonstrated in Figure 5 (a).

**City Complementation.** Our model can accomplish city complementation tasks without the need for additional retraining. We employ the same sliding window approach as in the infinite generation process. Figure 5 (b) visually showcases our model's capability in this task.

**Generation Based on Road Network.** Our model is highly adaptable for conditional generation tasks. In scenarios where we aim to generate a city layout based on a given road network, since the road network can be represented by a collection of polylines, we can employ the same encoder structure to encode the road network data. Then, we can concatenate this condition embedding with the mask tokens or directly append the road network embedding sequence to the end of the building sequence. In our implementation, we choose the latter approach to give our model the capacity to capture mutual attention between roads and buildings.. The results of this road conditional generation are illustrated in Figure 5 (c).

**2.5D Generation.** Recently, numerous efforts have aimed to generate infinite 3D scenes using data sources like RGBD images or satellite imagery (Lin et al., 2023; Chen et al., 2023; Raistrick et al., 2023). Additionally, some studies have focused on generation consistency (Shen et al., 2022). In this context, our model are capable of generate coherent infinite 3D cityscapes devoid of any conditions but with simple 3D building structure. To accomplish this task, we extended our output to include an additional height dimension. The outcomes of this extension are depicted in Figure 5 (d).

**Classification Task.** For the classification task, we compiled an extensive dataset encompassing urban layouts from cities worldwide, including Tokyo, Berlin, New York, and others. This dataset included 1,383,384 buildings, 300,000 blocks, and 15 distinct districts, with each district contributing 20,000 blocks. After training our model on this expanded dataset, we utilized the first-stage encoder to create a classification model capable of distinguishing these 15 districts. In the fine-tuning phase, our pretrained model achieved an impressive 95% accuracy within just one hour. Furthermore, by employing linear probing (adding a simple linear layer to the encoder's output), we achieved an 84% accuracy. In contrast, using a transformer with the same structure but without prior pretraining required 8 hours to reach a 90% accuracy rate. This highlights the superior efficiency and effectiveness of our pretrained model in the downstream classification task.

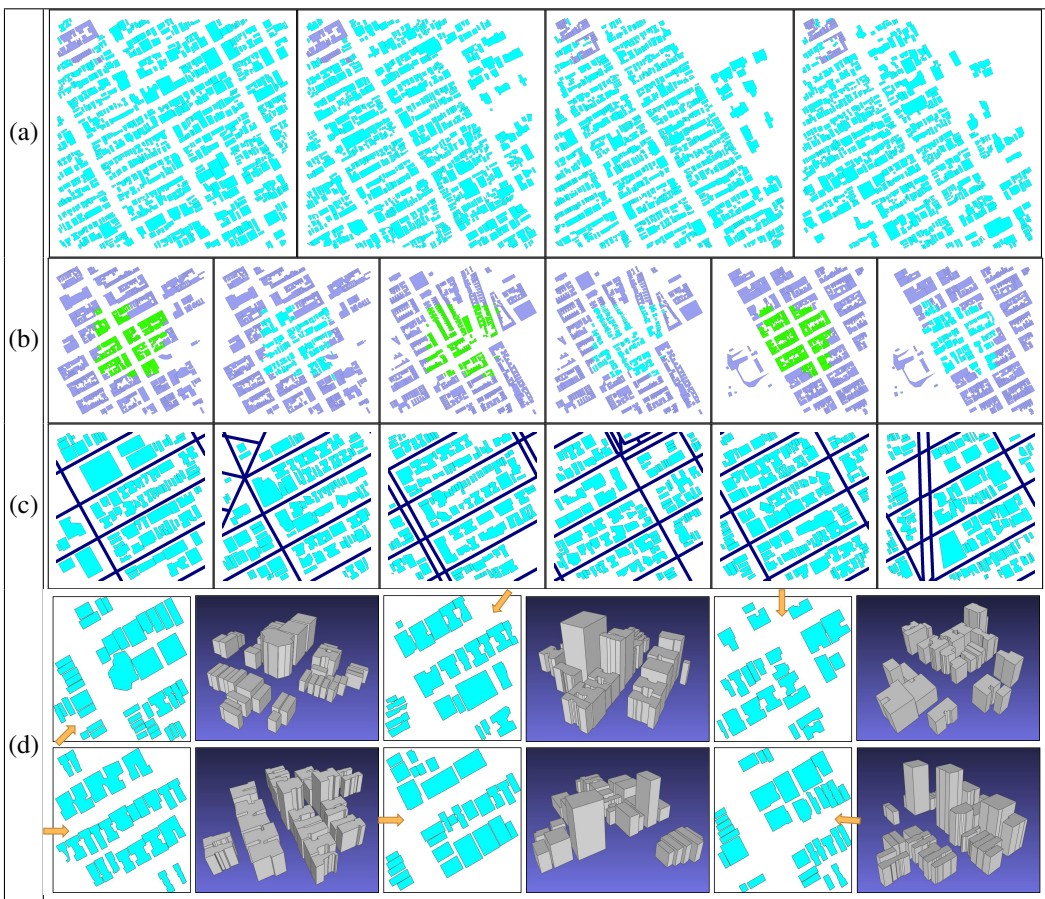

Figure 5: **(a) Results of Infinite Generation.** The buildings represented in purple denote the prior buildings, while the blue are the buildings generated by our model. **(b) Results of City Complementation.** In each pair, purple buildings represent the conditional city segment, while green and blue buildings represent the ground truth and generated buildings, respectively. **(c) Results Conditioned on Road Network.** In the images, the dark blue lines delineate the conditional road network, while the blue represent the generated buildings. **(d) Results of 2.5D City Generation.** Within each pair of images, the image on the right showcases the generated footprints layout. The orange arrow indicates the direction of observation, which corresponds to the 3D scenes displayed on the left.

## 6 CONCLUSION AND FUTURE WORK

We have introduced a method to generate city layouts of arbitrary building shapes in vectorized data. Our approach is effective for generating realistic city layouts, showing flexibility in various scenarios like infinite city generation, city complementation, and road-conditioned generation. Moreover, our training pipeline not only yields a generation model but also offers a meaningful representation of city layouts. Our results demonstrate superior performance compared to previous methods (Ha & Eck, 2017; Jyothi et al., 2021; Gupta et al., 2021; Han et al., 2023; Inoue et al., 2023), and our user study indicates that the results generated by our method are comparable to real-world data.

However, our approach does have certain limitations. Firstly, the iterative generation inference process can be time-consuming. Secondly, our model lacks a latent space for complete generation sampling, making it challenging to apply style transfer or interpolation within our model. Thirdly, there is still room for improvement in aligning the building and road widths in our results.

As a future direction, our model can be extended to various conditional generation tasks, as road conditioning has already been implemented. It holds promise to incorporate additional conditions like terrain, satellite imagery, and more. Additionally, our model architecture provides an opportunity to extend our 2D layout generation to the creation of realistic 3D city scenes.

## 7 REPRODUCIBILITY STATEMENT

To make all our experiment results reproducible, we submit codes with hyperparameters used for each method on each task in the supplementary material. We also provide an instruction on how to use our code to reproduce the experiment results.

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

## A    MODEL ARCHITECTURE

In our detailed model structure design, our model comprises the following components: a polygon embedding, an encoder, a Stage I decoder, a Stage II decoder, and an autoregressive model. All of these components are implemented using transformer architectures, and the specific parameter settings can be found in Table 5.

|  | layers | hidden | heads | mlp_ratio |
|---|---|---|---|---|
| Polygon embedding | 1 | 512 | 8 | 1 |
| Encoder | 12 | 512 | 8 | 4 |
| Stage I decoder | 3 | $16 \times 25^{\dagger}$ | 8 | 4 |
| Stage II decoder | 8 | 512 | 8 | 4 |
| Autoregressive model | 1 | $512 + 256^{*}$ | 8 | 1 |

Table 5: **Specific Model Architecture Parameters.** The symbol $\dagger$ indicates that each position has a dimension of 16, and there are a total of 25 positions in each patch. The symbol $*$ denotes that the latent representations generated by the decoder have a dimension of 512, while the input embedding to the autoregressive model has a dimension of 256.

A more detailed model design of our base model, the model capable of generating 2.5D cities, and the model that can generate city layouts based on the road network can be found in Figure A.1.

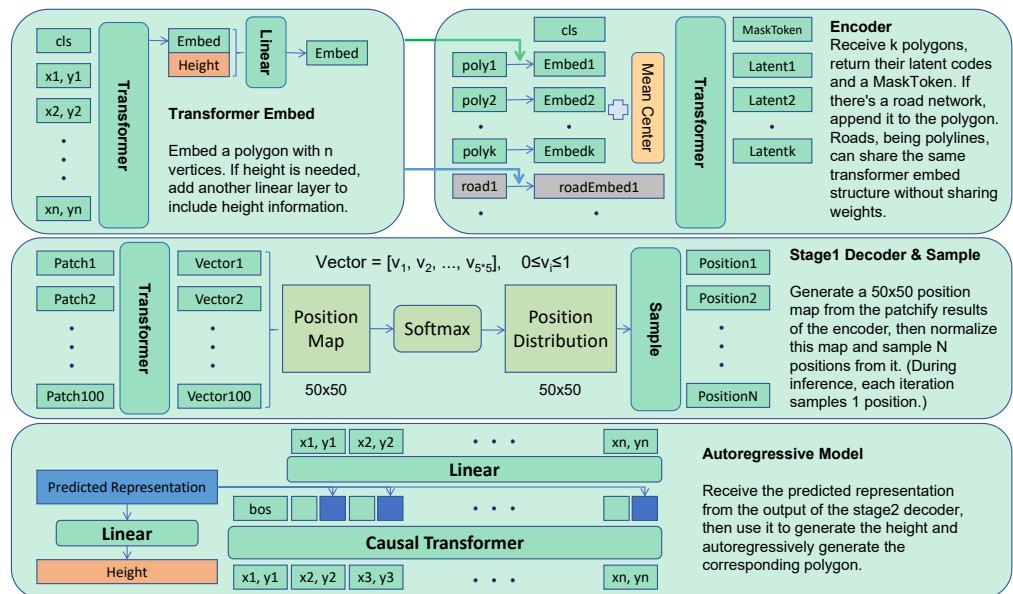

Figure A.1: **Detailed Structure in CityGPT.** From top to bottom, there is the detailed model structure of the encoder used in both Stage I and Stage I, the decoder of Stage I, the subsequent sampling process, and the autoregressive generation part at the end of Stage II. The overall structure can be found in Figure 2.

For the classification task mentioned in Section 5, the specific model structure can be found in Figure A.2. Note that the encoder part in the classification model has the same structure as the encoder part in CityGPT (in both Stage I and Stage II). We have outlined three phases with different training settings in Section 5. Firstly, in the fine-tuning setting, initialize the encoder with the trained Stage I encoder of CityGPT, then fine-tune the entire model. Secondly, in the linear probing phase, fix the encoder part initialized by CityGPT and only train the decoder. Thirdly, in training from scratch, randomly initialize all parts and then train the entire model.

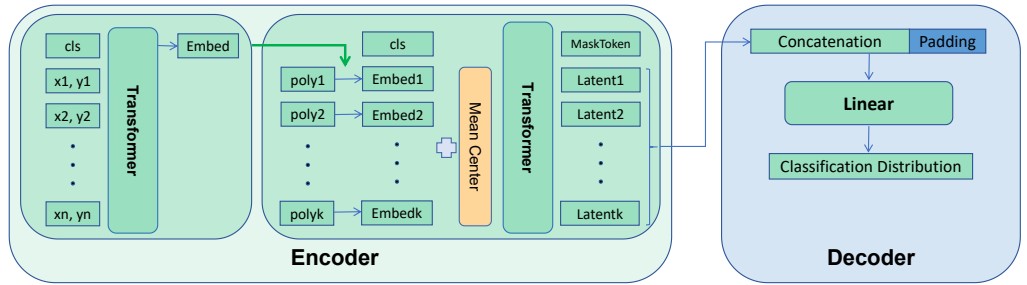

Figure A.2: **Classification Model:** The encoder part is the same as the encoder in CityGPT, and the decoder consists of a simple linear layer.

## B  DETAIL OF INFERENCE

The inference pipeline can be shown as the following Algorithm 1. Please note that in this algorithm, $\leftarrow$ denotes sampling from the corresponding distribution. As described in Section 3.4, in most cases (with a probability of 0.9), we determine the next position by selecting the one with the highest probability. We also discard positions that would generate polygons overlapping with previous polygons (otherwise, the overlap ratio is approximately 7%)

---

**Algorithm 1** Inference of CityGPT

---
**Output:** A set of building footprints $\mathbf{B}$.
 1: Uniformly sample initial position $\mathbf{P}_0$ from $[0, 50] \times [0, 50]$.
 2: Obtain initial footprints $\mathbf{F}_0$ using Stage II.
 3: $\mathbf{B}.\text{append}(\mathbf{F}_0 \leftarrow \mathcal{P}_2(\mathbf{F} \mid \emptyset, \mathbf{P}_0))$
 4: Obtain position map $\mathbf{M}$ using Stage I.
 5: $\mathbf{M} = \mathcal{P}_1(\bar{\mathbf{P}} \mid \mathbf{B})$
 6: Mask positions with buildings in $\mathbf{M}$ to be 0.
 7: $t = 1$
 8: **while** $\min(\mathbf{M}) > 0.5$ and $t < \text{max\_len}$ **do**
 9:     Mask positions with buildings in $\mathbf{M}$ to be 0.
10:     **while** $\min(\mathbf{M}) > 0.5$ **do**
11:         **if** $\text{Uniform}([0, 1]) < 0.9$ **then**
12:             $\mathbf{P}_t \leftarrow \text{Softmax}(\mathbf{M})$
13:         **else**
14:             $\mathbf{P}_t = \max(\mathbf{M})$
15:         $\mathbf{F}_t \leftarrow \mathcal{P}_2(\mathbf{F} \mid \mathbf{B}, \mathbf{P}_t)$
16:         **if** $\mathbf{F}_t$ intersects with $B$ **then**
17:             $\mathbf{M}(\mathbf{P}_t) = 0$
18:         **else**
19:             $\mathbf{B}.\text{append}(\mathbf{F}_t)$
20:             **Break**
21:     $\mathbf{M} = \mathcal{P}_1(\bar{\mathbf{P}} \mid B)$
22:     $t = t + 1$

---

We also conducted runtime performance analysis on our model inference pipeline, using the inference of LayoutTrans (Gupta et al., 2021) under the same computational conditions (single RTX3090) as a basis for comparison. We chose LayoutTrans for comparison because it generates the entire city layout autoregressively, similar to our inference setting where we generate buildings one by one iteratively, resulting in similar iteration steps during inference. The results are shown in Table 6. In this table, Our-Intersection indicates the case where we perform generation without the intersection validation mentioned above. These results reveal that the main reason for the increase in our inference time is due to the intersection detection and resampling parts. According to our experiments, the intersection detection part (Steps 16-17 in Algorithm 1) introduces a 0.8 times slowdown, while the

resampling operation (if intersect, then resample) introduces about a 1.0 times slowdown (meaning, on average, the 10th row in Algorithm 1 will cycle twice).

| | LayoutTrans | Ours-Intersection | Ours-Intersection (2.5D) | Ours |
|---|---|---|---|---|
| Time $(s)$ | 2.5 | 2.4 | 2.5 | 9.4 |

Table 6: The first row shows the models and corresponding tasks, and the second row displays the mean running time for generating a single city block layout.

## C    DETAIL OF CITYSIM

Inspired by DocSim (Patil et al., 2020), we designed CitySim to evaluate the similarity between pairs of city layouts. We treat each pair of city layouts $(B_1, B_2)$ as a bipartite graph, where each node represents the polygon footprints in the layout. Then we define the edge weight between two nodes as:

$$W(f_1^i, f_2^j) = \sqrt{\text{MinArea}(f_1^i, f_2^j)} \cdot 2^{-\Delta_e(f_1^i, f_2^j) - C_a(\sqrt{\text{Area}(f_1^i)} - \sqrt{\text{Area}(f_2^j)})},$$

where $f_1^i \in B_1$ and $f_2^j \in B_2$. $\text{MinArea}(f_1^i, f_2^j)$ represents the smaller occupancy area of these two footprints. $\Delta_e(f_1^i, f_2^j)$ represents the Euclidean distance between the mean centers of these two footprints. In (Patil et al., 2020), they chose $C_a = 2$; however, in our setting, the polygons are relatively smaller than the boxes in their setting compared to the whole canvas. Therefore, we choose $C_a = 20$ to ensure that both terms in the exponent are meaningful to the weight. Our experiments show that regardless of changing $C_a$ from 0 to 50, our model consistently outperforms other models. Thus, choosing $C_a = 20$ is not a painstaking decision. Then we find a maximum weight matching $M(B_1, B_2)$ in this bipartite graph using the well-known Hungarian method (Kuhn, 1955). So the similarity between two city layouts can be defined as:

$$\text{Sim}(B_1, B_2) = \frac{1}{|M(B_1, B_2)|} \sum W(f_1, f_2),$$

where the sum is over all pairs $(f_1, f_2) \in M(B_1, B_2)$. Then we define CitySim as:

$$\text{CitySim}(B \in \text{GEN}) = \min_{B_{gt} \in \text{GT}} (\text{Sim}(B, B_{gt})),$$

where GEN is the set of city layouts generated by our model, and GT is the set of groundtruth city layouts.

## D    DISCUSSION OF OVERFITTING

Inspired by the "Novel" score and "Unique" score in SkexGen (Xu et al., 2022a), our goal is to employ a similar score to demonstrate that our model isn't overfitting to the training dataset. However, since there are never two layouts exactly the same (given that we generate layouts in a continuous space), we prefer using pixel similarity between two sets of layouts to assess these scores. We define **PixelSim** as the same pixel ratio (same pixel/total pixel) between two layouts. The rendered images used to compute this similarity are at a pixel resolution of $500 \times 500$, with buildings represented in blue and edges in black.

We compute the similarity score between any two samples from the training sets (Train-Train), between any two samples from the generated set (Gen-Gen), and between any two samples from the training set and the generated set respectively (Gen-Train). Specifically, in the Gen-Train case, for each layout in the generated set, we identify the layout in the training set (comprising 32,000 layouts) with the highest PixelSim score (most similar) and assign that highest score to it. The mean-PixelSim is then calculated as the mean score over these 1000 generated samples. Then in the Gen-Gen case, the score for each generated layout is computed by finding the most similar layout in the remaining 999 generated layouts. To ensure fairness between the Gen-Train case and Train-Train

case, we randomly sample 1000 samples from the training set. For each of them, we find the most similar layout in the remaining 31,999 layouts.

The results are presented in Table 7 and Figure D.1. It's important to note that Gen-Gen serves as a metric to evaluate the diversity of generation results, while Gen-Train serves as a metric to evaluate the novelty of generation results. For comparison purposes, we provide the same scores for 1000 sampled training data (Train-Train), which serves as a baseline for comparison. It's worth mentioning that our rendered layouts are very dense images with only three types of pixels (white, blue, and black), so a 90% similarity at the pixel level still maintains a considerable distance between layouts.

|  | Train-Train (Baseline) | Gen-Gen (Uniqueness) | Gen-Train (Novelness) |
|---|---|---|---|
| mean-PixelSim (%) | 63.8 | 50.7 | 53.5 |

Table 7: The value from left to right represents the pixel similarity score between the training set and the training set, between the generated set and the generated set, and between the generated set and the training set, respectively.



Figure D.1: **Distribution of the PixelSim Score Among 1000 Samples.** From left to right are the distributions of PixelSim in the Train-Train case, Gen-Gen case, and Gen-Train case, respectively.

# E    IMPLEMENTATION DETAIL OF BASELINE

**SketchRNN** (Ha & Eck, 2017): We modify the datasets to fit SketchRNN in the following way: we consider each data block as a picture that SketchRNN needs to draw. Each time it finishes drawing one polygon inside the block, the pen lifts up from the canvas and moves to another place to start drawing the next polygon. During testing time, we modify the last line to directly connect to the start point of each polygon, thereby making all polygons closed shapes.

**AETree** (Han et al., 2023): We have improved the original AETree model so that it can not only generate buildings of simple box shapes. Specifically, we added an MLP network to the end of the AETree model, which transforms the simple box into a polygon with multiple vertices. Since the number of vertices in the polygon is not fixed, during the training process, in addition to using the L1 loss function to constrain the predicted polygon shape, we also adopted the BCE loss function to constrain the number of predicted polygon vertices.

**LayoutVAE** (Li et al., 2019): We customized the LayoutVAE model to suit our specific tasks by incorporating our polygon embedding class before the embedding layer and an autoregressive generation class after the final output layer. Notably, in our context, we only have a single class. As a result of these adaptations, the modified LayoutVAE is capable of generating the entire building layout from scratch.

**LayoutTrans** (Gupta et al., 2021): We have directly adapted the structure of the LayoutTransformer, but applied it to our specific dataset. Our initial step involved arranging all polygons within each block in ascending order based on the x-coordinate of their mean center. Subsequently, we flattened all these polygons into a single long sequence, adding four extra tokens corresponding to the start token ($S$), terminal polygon token ($T$), end token ($E$), and pad token ($P$). And we have also discretized the polygon coordinates in line with the original paper, employing a quanti-

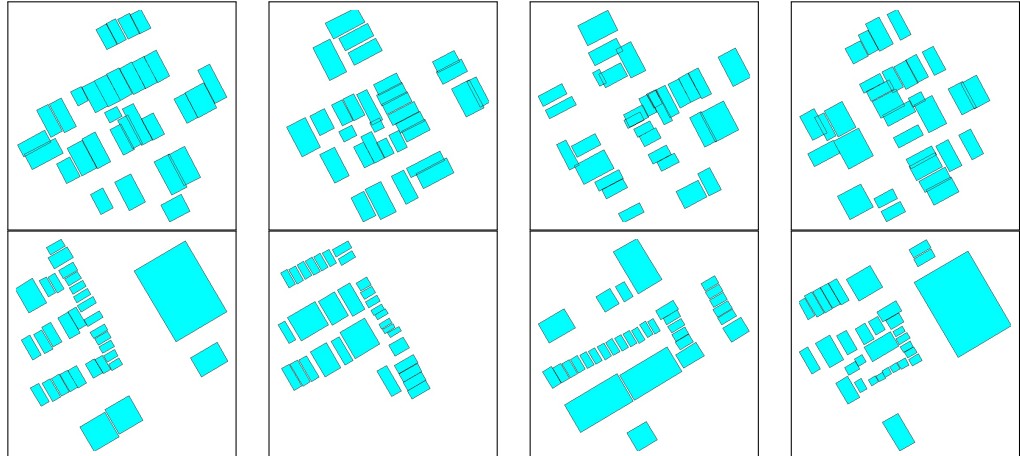

Figure F.1: **Qualitative Comparison of Ablation Study in Box Setting:** The first row displays the results of our model with a graph structure, while the second row showcases our final configuration.

zation approach with a precision of unit8. Then we represent each block in the following format: $(S, x_1^1, y_1^1, \ldots, x_1^{n_1}, y_1^{n_1}, T, x_2^1, \ldots, x_N^{n_N}, y_N^{n_N}, E)$.

**LayoutDM** (Inoue et al., 2023): We have made adaptation to the primary structure of LayoutDM, except for some techniques that are not applicable to our task setting, such as Modality-wise diffusion and Adaptive Quantization. We experimented with two flattened sequence settings to integrate their model into our task. The first sequence setting mirrors the adaptation of LayoutTransformer as described above. In the second sequence setting, we padded each polygon to 40 dimensions since our maximum vertex count per polygon is 20. So the second representation sequence is as follows: $(x_1^1, y_1^1, \ldots, x_1^{n_1}, y_1^{n_1}, Pad, Pad, \ldots, x_2^1, \ldots, x_N^{n_N}, y_N^{n_N}, Pad, \ldots, Pad)$. And we found that the second setting yields relatively better results. The subsequent results are based on this second sequence setting.

# F  MORE ABLATION STUDIES

|  | Rcon. L1 $\downarrow$ | Rcon. Giou $\downarrow$ | FID $\downarrow$ | WD $\downarrow$ $(area)$ |
|---|---|---|---|---|
| Patch with matching | 0.17 | 0.74 | $--$ | $--$ |
| Patch with chamfer | 0.20 | 0.78 | $--$ | $--$ |
| Graph embedding | 0.095 | 0.47 | 98.9 | 0.0407 |
| Ours | **0.082** | **0.42** | **83.2** | **0.0158** |

Table 8: **Quantitative Results of Ablation Study in Box Setting:** We present four metrics for comparing different model structures. The initial two metrics correspond to the reconstruction loss, specifically L1 and Giou. The subsequent two metrics, FID and WD, are as described in section 4.2. It is evident that our final configuration outperforms the others across all metrics.

In this section, we delve into a series of experiments involving a simpler task aimed at generating box layouts. These layouts encompass 32 boxes, each representing the bounding box of a building's footprint. These bounding boxes are defined by 5 dimensions: $b_i = (x, y, h, w, angle)$, capturing the box's central coordinate, height, width, and main axis angle.

To achieve flexible generation choices and consistent infinite generation, we design a reconstruction task describe in section 3 which is our main idea. Then the remaining question is to design a model based on part of the existing layouts and then generate the complement layout. Some work (Kong et al., 2022; Gupta et al., 2021) achieve this complement task by pre-define the order of each box in the layout. but this order will constrain the generation choice in many cases, such as complementing a central part of a city as described in section 5. So without this predefined order, the main challenge

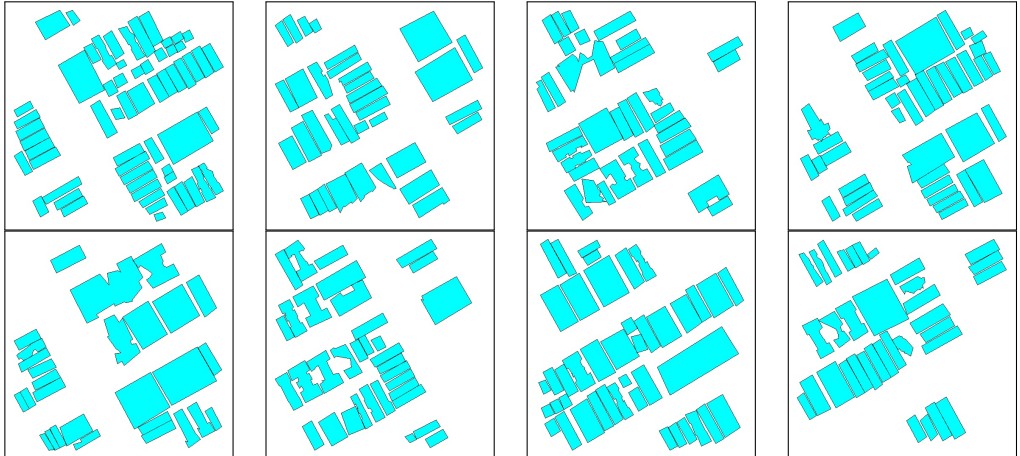

Figure G.1: Results of generation in a discretized situation that supports sampling in polygon generation based on given positions.

in the task is to define a correspondence between the ground-truth layout boxes and the target layout boxes.

In our model, we handle this correspondence by introducing a position prediction part, then we can use the position to build a correspondence between the target layout and generate layout. And also try some other way to build this correspondence. The first is directly finding a match with the minimal distance, this is an idea of the DETR (Carion et al., 2020), and similar to the famous distance metric Chamfer Distance has been widely used in point cloud generation. But unfortunately, both of them can not converge to our desired pattern. We think it may be caused by the high dimensionality and relative sparsity in our dataset.

The second solution is considering constructing a graph structure $G(V, E)$ for the layout.

$$V = \{b_i\} = \{(x_i, y_i, h_i, w_i, angle_i)\}$$

Where each node corresponds to a box.

$$E \in \{e_1, e_2, e_3, e_4\}$$

Our edges contain four types. and for each node (box) there are at most four out edges with correspond to four different types. Each type connects to the certain box that is most near to the node in axis $-x, +x, -y, +y$. And we have an easy way to represent this graph structure by assigning each node with extra two dimension information with respect to its orders in the $x$ and $y$ axis. Then if the order is seen to be the position with a certain constraint that no two positions have the same $x$ or $y$ coordinate, we can use the same model structure to fit this setting. Although the results show that the graph construction can give us sensible results, it suffers from the setting with arbitrary node numbers. The quantitative comparison and qualitative comparison are shown in table 8 and figure F.1. Based on these results, it is evident that our model with a graph structure acting as a correspondence link experiences a decrease in the diversity of the generated boxes.

## G  POLYGON GENERATION WITH SAMPLING

We can readily adapt our model to the discretized situation, enabling us to perform sampling in polygon generation based on given positions. To achieve this, we employ a discretization approach wherein we divide our 500-meter block into a 250 by 250 mesh. Within this mesh, we map each continuous vertex of the polygons into the grid of the mesh. In cases where two vertices of a particular polygon map to the same grid cell, we merge these vertices. Subsequently, we retrain our model and replace the L1 loss from the original setup with cross-entropy loss. This modification allows us to obtain favorable generation results, as demonstrated in Figure G.1.

In this case, we can use beam search to get the top k possible city layouts given the same positions as shown in figure G.2. We can discover that the difference between different samples from beam

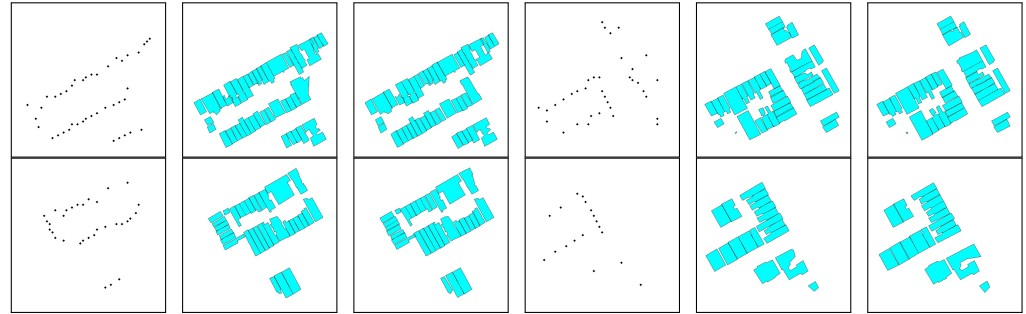

Figure G.2: Results of sampling different city layouts from a given set of positions. In each set, the left image showcases the position set, while the right two images showcase the results generated from the same position set.

search is little, which can certify the quality of our method in using the $\delta$ function to approximate the probability of the second stage in section 3.

## H    MORE RESULTS

The comparison between ground-truth and ground-truth with minor noise is shown in Figure H.1. The breakdown visualization of a single layout generation process is shown in Figure H.2.

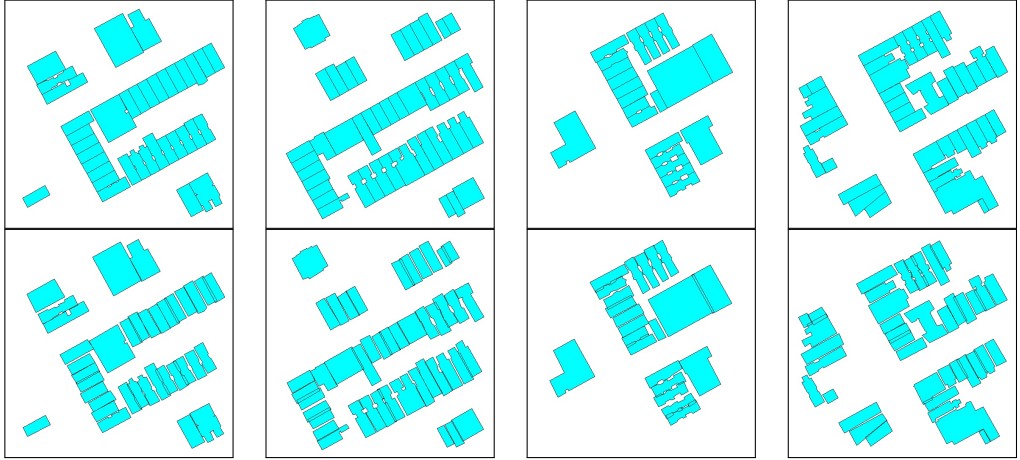

Figure H.1: **Comparison between different settings of the user study.** The first row shows the ground-truth samples, and the second row shows the ground-truth samples with minor noise.

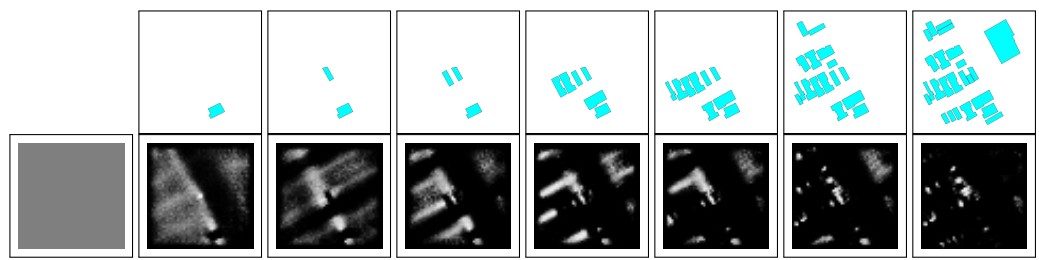

Figure H.2: The first row shows the generated layout at a certain timestamp, and the second row shows the predicted positional map at these corresponding timestamps. From left to right are the timestamps of 0, 1, 2, 3, 6, 9, 16, and 24.

More visualized results are shown in Figure H.3, H.4, H.5, and H.6.

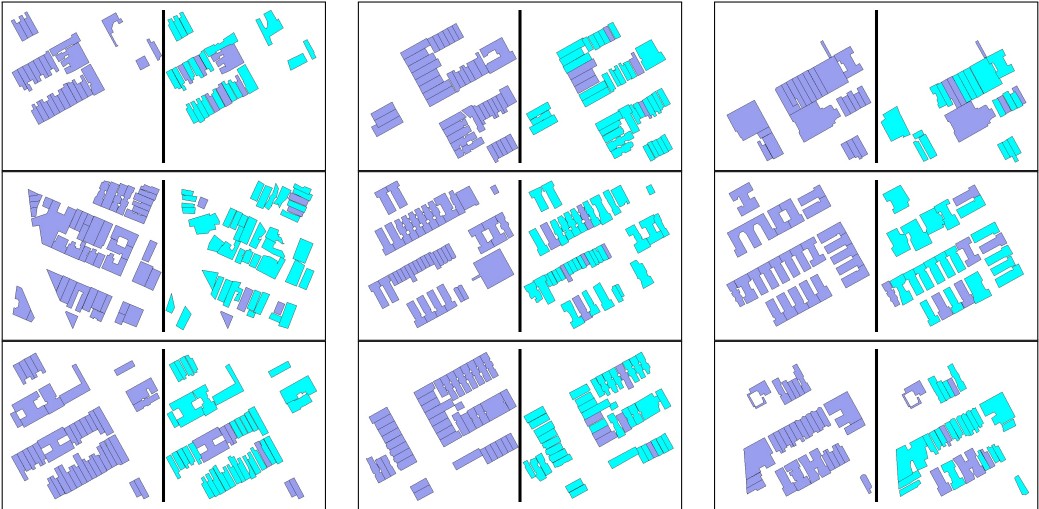

Figure H.3: **Results of Reconstruction for the Second Stage:** For each pair, the left side depicts the ground truth building layout, while the right side displays the reconstruction result. In the reconstruction, purple represents the remaining buildings, and blue represents the predicted buildings.

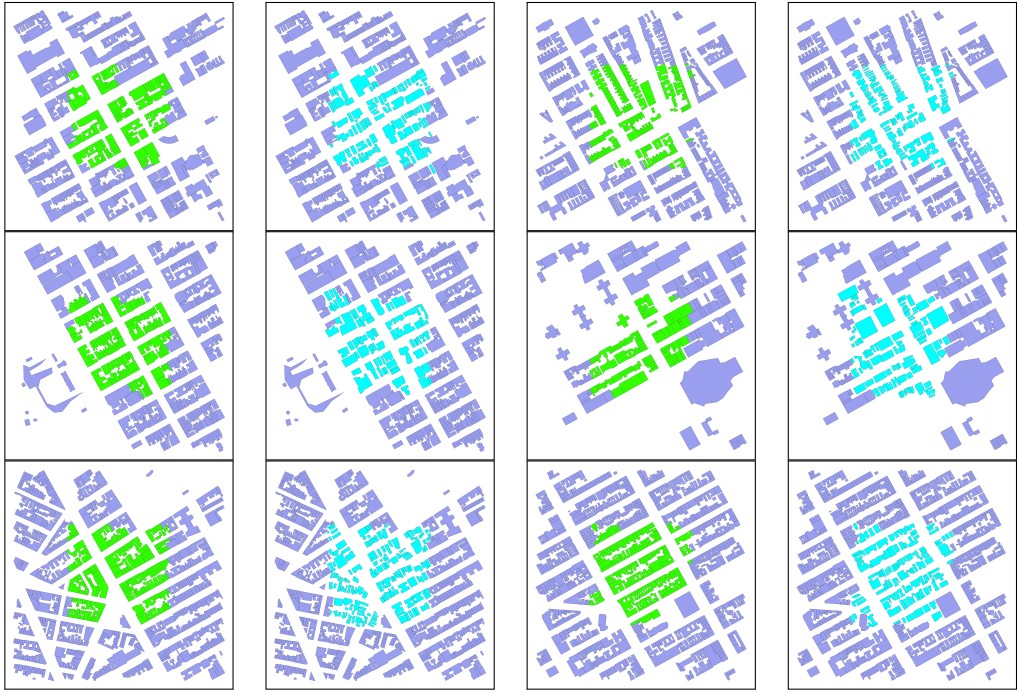

Figure H.4: More Results of City Complementation.

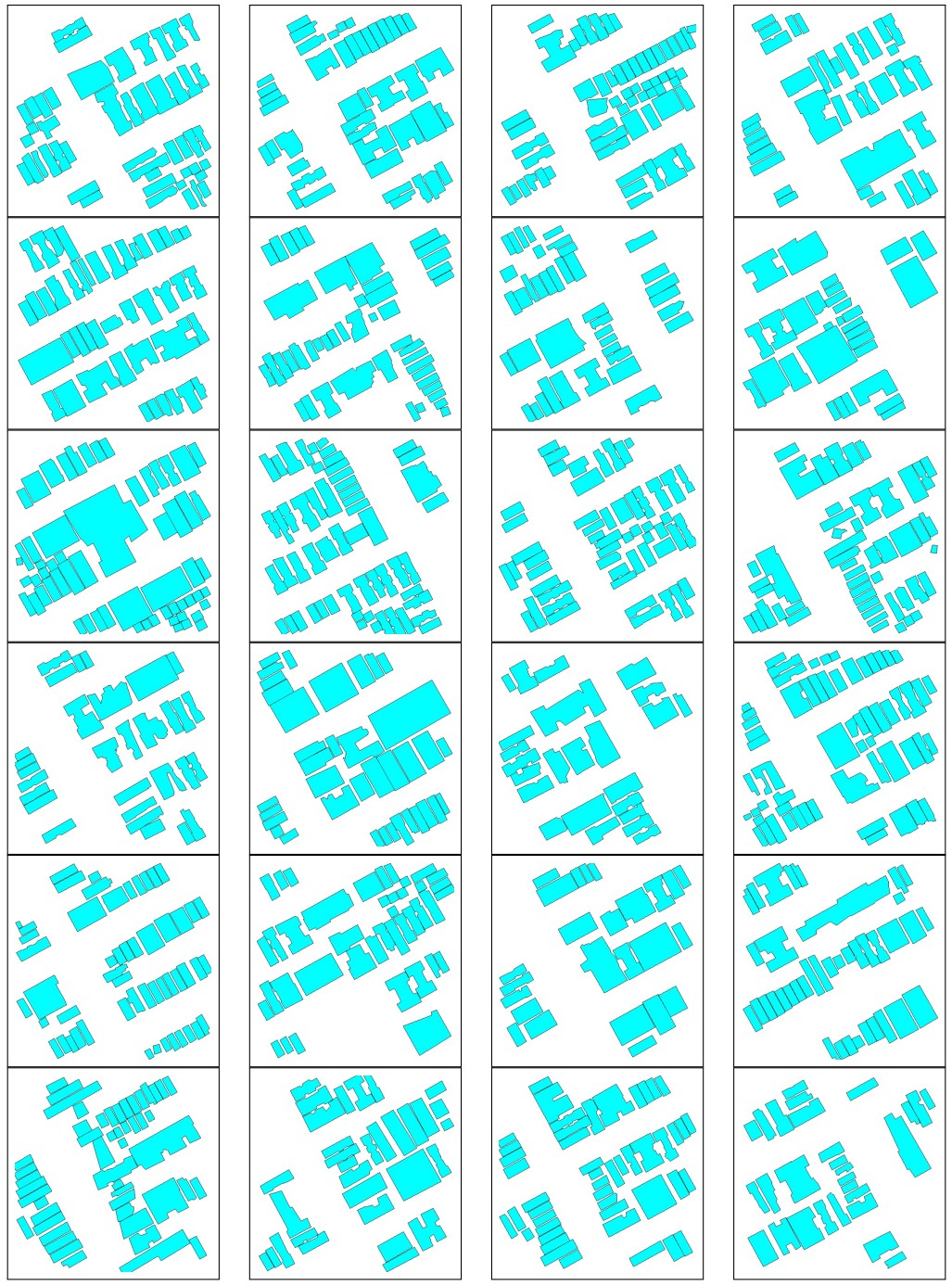

Figure H.5: More Results of Generation.

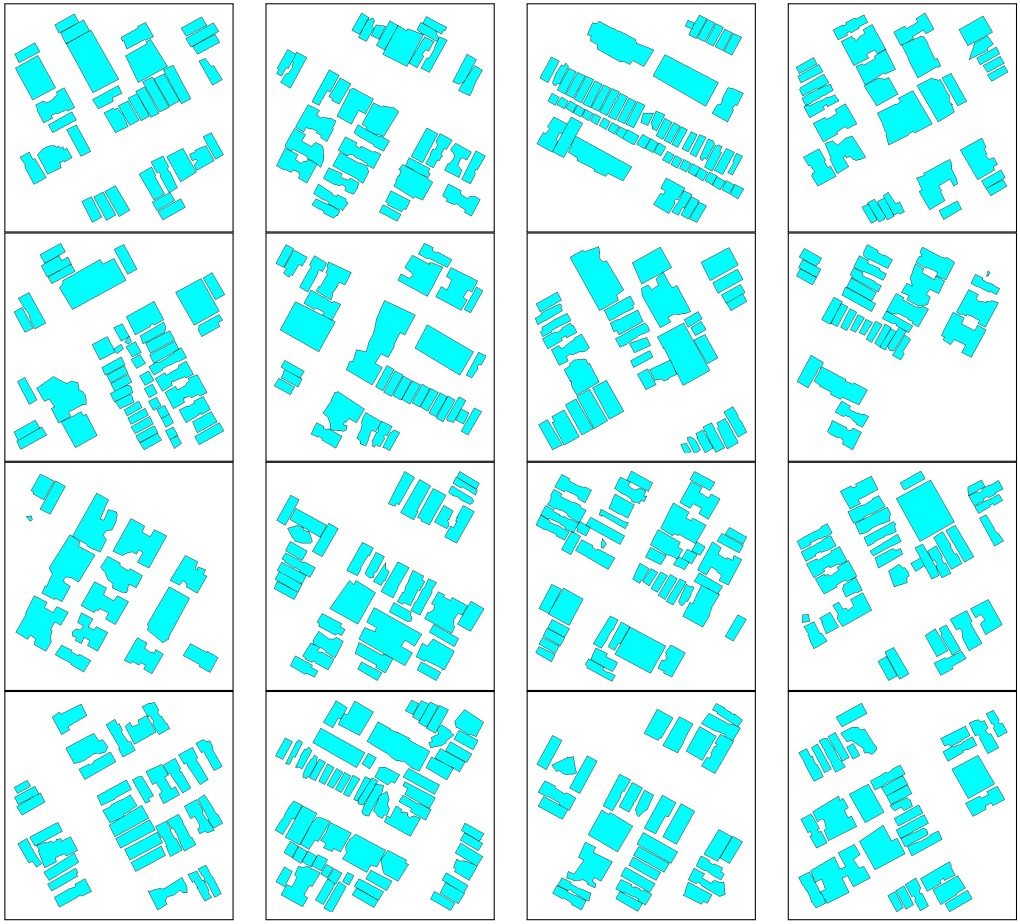

Figure H.6: More Results of Generation Trained with Random Flip.

