# OpenReview forum: "CityGPT: Generative Transformer for City Layout of Arbitrary Building Shape"
_ICLR.cc/2024/Conference — Submitted to ICLR 2024_

### Official Review · Reviewer_K6Hc · 2023-10-29

**Soundness:** 3 good
**Presentation:** 3 good
**Contribution:** 3 good
**Rating:** 6
**Confidence:** 4

**Summary:**

Authors proposed a two-stage transformer-based generative model for modeling city layout with arbitrary polygon building shape. The pipeline first generate center position for each building, and then autoregressively generate the polygon shapes of the building. The model is trained in a MAE fashion, and at inference time iteratively generates the masked building position and shape conditioned on existing unmasked ones. Results demonstrate the effectiveness of this two-stage approach. Further experiments on generating buildings with extruded height are also shown for completeness.

**Strengths:**

Writing is clear and easy to understand. The two-stage approach with MAE-like training is novel and breaks down a hard problem into iterative generation of layout and geometry. The autoregressive transformer is also more capable as it can generate arbitrary polygon shapes as opposed to just the 2D bounding boxes. Results are very extensive including ablation studies of the two-stage versus one-stage. Overall I am satisfied with the quality and novelty of this work.

**Weaknesses:**

Section 3.4 inference stage needs more detailed explanation. It is not very clear how the full building blocks are generated unconditionally from nothing. Some important evaluation metrics are also missing. A large autoregressive transformer (e.g. 12 layers) is prone to overfitting to the training set. Since the generated output are vector sequences, it should be easy to evaluate the novelness and uniqueness scores as in SkexGen (Autoregressive generation of CAD construction sequences) or CurveGen (Engineering sketch generation for computer-aided design). That way we will know the model is not simply remembering the training set.

**Questions:**

I would appreciate if authors can explain a bit more details about how the inference stage is conducted. Authors should also provide some proof that the trained model is not over-fitting to the training set. Novel and Uniqueness scores are a good benchmark as I mentioned. Given a generated result, authors can also illustrate the nearest neighbour search from the training set and compare their similarity.

---

> ### Author Response · Authors · 2023-11-22
> **Add detailed explainations of inference pipeline, novelness score, and uniqueness score.**
>
> Thank you for the constructive feedback, and we provide some discussion and responses regarding your comments.
>
> 1. “Section 3.4 inference stage needs more detailed explanation. It is not very clear how the full building blocks are generated unconditionally from nothing. Some important evaluation metrics are also missing. A large autoregressive transformer (e.g. 12 layers) is prone to overfitting to the training set. Since the generated output are vector sequences, it should be easy to evaluate the novelness and uniqueness scores as in SkexGen (Autoregressive generation of CAD construction sequences) or CurveGen (Engineering sketch generation for computer-aided design). That way we will know the model is not simply remembering the training set.”
>
> **Answer:** Thanks for your constructive comments. We have added a detailed explanation of our inference pipeline in Appendix B. Additionally, to better evaluate our model's generation results, we introduced a new metric called CitySim, inspired by READ[1]. You can find the results in Table 2, and the specific design of these metrics is detailed in Appendix C. Finally, thanks for your suggestion regarding the evaluation of novelty and uniqueness scores. We added some discussion and scores in Appendix D to clarify that our model is not simply remembering the training set.
>
> [1] Patil, Akshay Gadi, et al. "Read: Recursive autoencoders for document layout generation." CVPR 2020.
>
> 2. “I would appreciate if authors can explain a bit more details about how the inference stage is conducted. Authors should also provide some proof that the trained model is not over-fitting to the training set. Novel and Uniqueness scores are a good benchmark as I mentioned. Given a generated result, authors can also illustrate the nearest neighbour search from the training set and compare their similarity.”
>
> **Answer:** Thanks for your constructive comments. We have modified our paper writing to meet your suggestions. Details about the inference stage can be found in Appendix B, and the proof of no overfitting can be found in Appendix D. Additionally, we introduced a new metric called CitySim, which can illustrate the nearest neighbor, to better evaluate our generated results.

---

### Official Review · Reviewer_zw2r · 2023-10-30

**Soundness:** 2 fair
**Presentation:** 2 fair
**Contribution:** 2 fair
**Rating:** 5
**Confidence:** 4

**Summary:**

The paper introduces CityGPT, a novel approach for generating city layouts without relying on prior information like satellite images or layout graphs. This model leverages transformer-based masked autoencoders to sequentially learn two conditional models: one for building center positions given unmasked layouts, and the other for masked layouts given sampled center positions and unmasked layouts. Additionally, CityGPT incorporates an autoregressive polygon model, enabling it to generate city layouts with diverse building footprint shapes. The results demonstrate significant performance improvements over baseline methods, and CityGPT proves versatile in various generation tasks, including 2.5D city generation, city completion, infinite city generation, and conditional layout generation.

**Strengths:**

1. The paper is well organized.
2. The experiments somehow proves the effectiveness of the proposed method.

**Weaknesses:**

1. Recent studies, such as InfiniCity and CityDreamer, have focused on creating city layouts, incorporating both roads and buildings. However, this particular work only generates buildings without roads, which may limit its practical applicability in real-world scenarios.
2. The paper is not clearly written, missing too many details in Sections 3.2 and 3.3. After reading the two sections, it is still unclear how to convert the "Predicted Position Map" to "Reconstructed Building Layout".

**Questions:**

1. What is "in f city layout generation" mentioned in the third contribution?
2. The first phase should undergo a comparison with InfiniteGAN, employed in InfiniCity, and MaskGIT, utilized in CityDreamer. Additionally, if feasible, it should be contrasted with Diffusion models, as all of these models are applicable to both inpainting and outpainting tasks. Furthermore, all three models have the capability to directly generate footprint masks. In comparison to these three models, what specific advantages does the proposed model bring?
3. It is unclear how to generate the height of the buildings. According to the definitions in Section 3.1, the buildings only contains the coordinates of footprints.

---

> ### Author Response · Authors · 2023-11-22
> **Add detailed explainations of model structure and inference pipeline.**
>
> Thank you for the constructive feedback, and we provide some discussion and responses regarding your comments.
>
> 1. “Recent studies, such as InfiniCity and CityDreamer, have focused on creating city layouts, incorporating both roads and buildings. However, this particular work only generates buildings without roads, which may limit its practical applicability in real-world scenarios.”
>
> **Answer:** We acknowledge your concern about the applicability of our model compared to existing models like InfiniCity or CityDreamer. However, it's important to note that we address a different problem compared to InfiniCity and CityDreamer. Both InfiniCity and CityDreamer first generate a 2D semantic map or depth image of the layout, then extend it to 3D voxel rendering, focusing on rasterized data. Our approach, on the other hand, focuses on vectorized data generation. In contrast to the image-represented layout generated by InfiniCity and CityDreamer, our layout comprises a set of polygons represented by arrays. In our task field, 3D buildings are also represented as arrays with additional height value, or in a more sophisticated manner, using wireframe representation. To the best of our knowledge, we believe that the work most closely related to our problem is Globalmapper [1], which also focuses on generating vectorized city layouts. It's important to note that vectorized representations are more concise than rasterized data. By tackling this challenge, we believe our approach offers a valuable exploration in the field.
>
> [1] He, Liu, and Daniel Aliaga. "GlobalMapper: Arbitrary-Shaped Urban Layout Generation." ICCV 2023.
>
> 2. “The paper is not clearly written, missing too many details in Sections 3.2 and 3.3. After reading the two sections, it is still unclear how to convert the "Predicted Position Map" to "Reconstructed Building Layout".
>
> **Answer:** We apologize for the unclear explanation in our paper writing. We have updated our Appendix A to better demonstrate our model pipeline. Additionally, we have added a detailed explanation of our inference pipeline in Appendix B.
>
> 3. “What is "in f city layout generation" mentioned in the third contribution?”
>
> **Answer:** We apologize for the typo in our writing. The correct version is "in city layout generation." We have rectified this error with blue text.
>
> 4. “The first phase should undergo a comparison with InfiniteGAN, employed in InfiniCity, and MaskGIT, utilized in CityDreamer. Additionally, if feasible, it should be contrasted with Diffusion models, as all of these models are applicable to both inpainting and outpainting tasks. Furthermore, all three models have the capability to directly generate footprint masks. In comparison to these three models, what specific advantages does the proposed model bring?”
>
> **Answer:** Admittedly, InfiniteGAN has shown its effectiveness in inpainting and outpainting tasks. However, for the following reasons, we are not including it in our baseline.
>
> Firstly, InfiniteGAN focuses on generating infinite-pixel images, while our task does not involve image generation; we aim to generate vectorized data, facing different challenges compared to image generation tasks.
>
> Secondly, our approach is not limited to application tasks similar to inpainting and outpainting in the image generation field. Thanks to the vectorized data representation, we can easily condition on some buildings and generate others (not just mask a certain area in the layout) or delete specific buildings from the layout. In image generation, it is challenging to handle a certain building independently, as operations usually focus on a pixel area containing several buildings or incomplete building structures.
>
> Due to the distinct problem setting and application scenarios between our vectorized data generation task and the rasterized data (image) generation task of InfiniteGAN, we are not including it in our baseline.
>
>
> 5. “It is unclear how to generate the height of the buildings. According to the definitions in Section 3.1, the buildings only contains the coordinates of footprints.”
>
> **Answer:** We apologize for the unclear explanation in our paper writing. We have updated our Appendix A and added some new figures to better demonstrate the further experiments, including the setting that generates a 2.5D city.

---

> > ### Comment · Reviewer_zw2r · 2023-11-22
> > **The comparison between vectorized and image-based representations**
> >
> > After reviewing the rebuttal, I still believe that it is crucial to compare vectorized-based and image-based representations.
> >
> > In road generation, there is a paper called "Neural Turtle Graphics for Modeling City Road Layouts", which uses vectorized-based representation. However, upon comparison with image-based representations like Pix2Pix and Diffusion, we observed significantly inferior results.
> >
> > Therefore, I am curious about whether the proposed method outperforms the image-based methods although it is more relative to  "GlobalMapper: Arbitrary-Shaped Urban Layout Generation".
> >
> > I will refrain from raising my rating until I see relevant experimental results.

---

> ### Author Response · Authors · 2023-11-23
> **Our experiments are adequately designed to validate the effectiveness of our method in the task of generating layouts.**
>
> We respectfully disagree with the necessity to compare our method with image-based generation methods for the layout generation task.
>
> Firstly, the generation of roads differs significantly from building layout generation. In terms of data characteristics, roads typically comprise irregular polylines that may follow winding paths. Conversely, building layouts are characterized by a regular arrangement of building polygons. Additionally, the representation of layout data in vectorized format requires an extra step to convert rasterized images into vectorized polygons when utilizing image-based representation. Importantly, this conversion process is often non-trivial (and not standardized) and introduces discretization artifacts, which may lead to lower-quality results compared to Level of Detail 2 (LoD2) data, as exemplified in CityDreamer. [1].
>
> Secondly, city layout generation is an established research area, and we adhere to a consistent experimental protocol shared by several related works such as GlobalMapper [2], LayoutDM [3], and LayoutFormer++ [4]. These studies exclusively employ vectorized representation baselines. As such, we contend that our experiments are adequately designed to validate the effectiveness of our method in the task of generating layouts.
>
> [1] Xie, Haozhe, et al. "CityDreamer: Compositional Generative Model of Unbounded 3D Cities." arXiv preprint arXiv:2309.00610 (2023).
>
> [2] He, Liu, and Daniel Aliaga. "GlobalMapper: Arbitrary-Shaped Urban Layout Generation." Proceedings of the IEEE/CVF International Conference on Computer Vision. 2023.
>
> [3] Chai, Shang, Liansheng Zhuang, and Fengying Yan. "LayoutDM: Transformer-based Diffusion Model for Layout
> Generation." Proceedings of the IEEE/CVF Conference on Computer Vision and Pattern Recognition. 2023.
>
> [4] Jiang, Zhaoyun, et al. "LayoutFormer++: Conditional Graphic Layout Generation via Constraint Serialization and Decoding Space Restriction." Proceedings of the IEEE/CVF Conference on Computer Vision and Pattern Recognition. 2023.

---

### Official Review · Reviewer_YWWm · 2023-10-30

**Soundness:** 3 good
**Presentation:** 3 good
**Contribution:** 3 good
**Rating:** 6
**Confidence:** 4

**Summary:**

This paper focuses on the city layout generation task. A generative pre-trained transformer, i.e., CityGPT, is proposed for modeling city layout distributions from large-scale layout datasets. The distribution of buildings’ center positions is first learned. The distribution of masked layouts is then learned based on the sampled center positions and unmasked layouts. The city layouts are represented as arbitrary shapes instead of boxes. The experimental results demonstrate the effectiveness of the proposed method on several generation tasks.

**Strengths:**

This paper is the first to represent layouts of arbitrary scales and shapes without any prior conditions. The proposed two-stage decomposition modeling approach for city layout can accomplish various layout generation tasks. The experimental results demonstrate superior performance compared to existing works.

**Weaknesses:**

1. The runtime performance analysis should be conducted, including both the 2D and 2.5D generation.
2. Several layout generation works [1-4] should be cited and discussed in the paper.
3. Some details in the further experiments are missing. For the classification task, the architecture of the classification model used here is unclear. For the 2.5D generation, the training details after adding the additional height dimension are unclear. For the generated based on the road network, why choose the latter approach, rather than concatenating the condition embedding with the mask tokens?
4. There are some typos in the paper. For example, “in f city layout generation” in the last paragraph of the introduction section. “our results demonstrate” in the first paragraph of the conclusion section.

[1] Zheng, Xinru, et al. "Content-aware generative modeling of graphic design layouts." ACM TOG, 2019.

[2] Zhang, Junyi, et al. "LayoutDiffusion: Improving Graphic Layout Generation by Discrete Diffusion Probabilistic Models." ICCV 2023.

[3] Jiang, Zhaoyun, et al. "LayoutFormer++: Conditional Graphic Layout Generation via Constraint Serialization and Decoding Space Restriction." CVPR 2023.

[4] Chai, Shang, Liansheng Zhuang, and Fengying Yan. "LayoutDM: Transformer-based Diffusion Model for Layout Generation." CVPR 2023.

**Questions:**

1. For the user study, why the results of the proposed model are more realistic than the ground-truth layouts?
2. Will the dataset be released to facilitate future research in the community?

---

> ### Author Response · Authors · 2023-11-22
> **Add runtime performance analysis, cite related works, and improve the paper writing.**
>
> Thank you for the constructive feedback, and we provide some discussion and responses regarding your comments.
>
> 1. “The runtime performance analysis should be conducted, including both the 2D and 2.5D generation.”
>
> **Answer:** Thanks for your constructive comments. We have conducted a runtime performance analysis on our inference stage, including both 2D and 2.5D generation, as detailed in Appendix B. Our method could achieve same level of inference efficiency as baselines.
>
> 2. “Several layout generation works [1-4] should be cited and discussed in the paper.”
>
> **Answer:** Thanks for your constructive comments. We have cited and discussed them in the revised Section 2. We have also updated Table 1 to include the discussion of a work from your recommended list.
>
> 3. “Some details in the further experiments are missing. For the classification task, the architecture of the classification model used here is unclear. For the 2.5D generation, the training details after adding the additional height dimension are unclear. For the generated based on the road network, why choose the latter approach, rather than concatenating the condition embedding with the mask tokens?”
>
> **Answer:** Thanks for pointing out the weak points in our paper writing. We have added detailed explanations of our CityGPT model structure and the model structure in further experiments in Appendix A. Additionally, we have updated the expression in Section 5 about the road condition generation. We chose the latter approach to generate the layout condition on the road network to enable our model to have the capacity to learn mutual attention between roads and buildings.
>
> 4. “There are some typos in the paper. For example, “in f city layout generation” in the last paragraph of the introduction section. “our results demonstrate” in the first paragraph of the conclusion section.”
>
> **Answer:** We apologize for the typo in our writing. The correct version is “in city layout generation” and “Our results demonstrate”.  We have rectified this error with blue text.
>
> 5. “For the user study, why the results of the proposed model are more realistic than the ground-truth layouts?”
>
> **Answer:** As mentioned in our user study description, we added some small noise to the ground-truth data, which could be a reason for these results. Initially, we added this noise because most of the ground-truth data were perfectly aligned, and we assumed it might provide a simple way to distinguish them. However, we now consider that it may cause confusion for readers. Therefore, we have provided new user study results in Section 4.2 in cases where no noise is added to the ground truth. Additionally, we offer visual comparisons between ground truth with noise and ground truth in Appendix H. Even without nosies, our method can still outperform human baseline.
>
> 6. “Will the dataset be released to facilitate future research in the community?”
>
> **Answer:** Some of our datasets are open source, and those that we construct ourselves will definitely be released. We will include all of them on our future project page.

---

### Official Review · Reviewer_2WzD · 2023-11-01

**Soundness:** 2 fair
**Presentation:** 2 fair
**Contribution:** 2 fair
**Rating:** 3
**Confidence:** 4

**Summary:**

This paper introduces a transformer-based generative model of city layouts. The model has two phases, based on masked autoencoders: the first phase learns to predict a probability distribution over likely locations for building centroids; the second phase takes the position information and autoregressively predicts the vertices of the buildings. At test time, the two phases are alternated to allow autoregressive sampling of a city layout. Experiment shows that the proposed method generates reasonable layouts, and outperforms prior works (either for this task or for more general layout generation) over multiple metrics, including a human perceptual study.

**Strengths:**

- An interesting problem that can probably provide some insight to the more general layout generation problem as well.
- Can enable a range of applications, as illustrated in the paper.
- Reasonable results that clearly work better than some of the baselines
- The two-stage pipeline where a location distribution is used to directly condition polygon generation is rather new.

**Weaknesses:**

- Not too much novelty: the idea of predicting a probability distribution over building, the idea of autoregressive generation of polygons and the idea of transformer based layout generation can all be traced to prior works. While the domain (city generation) and the combination of techniques are novel (conditioning polygon generation directly on the location distribution), such novelty are likely not directly useful for people who are not interested in this particular problem. Subsequently, results quality would matter much more and
- The evaluation is underwhelming. Most baselines are for general layout problems that has much weaker constraints than the specific problem. The one baseline that addresses this problem (AETree) is a weak one the doesn't even compete with more general methods. Even with this set of baselines, I am not sure whether the proposed method really generates better layout, based on the qualitative results. The quantitative metrics are too generic (FID is too general for evaluating such layout visualizations, WD over edge/area/ratio doesn't really evaluate layout quality), the user/perceptual study is also not well conducted (ground truth shouldn't have minor noises, visualizations of layout should be better so humans can actually judge if the layouts are good i.e. not just with blue boxes/meshlab screenshots).
- Lack of evaluation over whether the method can generate novel and diverse layouts that are different from the training set. I am not convinced that the model is not overfitting to some training samples.
- The problem setting isn't particularly useful: without streets, roads, building types and other city elements, I can't see how this can be helpful in any real city planning / modeling tasks. I am also not sure whether representing buildings as 2D polygon contours adds much over just specifying the location and size of buildings: one would need to model the 3D building in some other ways anyways.
- A few technical issues that need to be addressed: see questions below.

**Questions:**

As mentioned in the weakness section, I have concerns over the evaluation protocol, providing more evidence that the model can generate realistic layout (i.e. with more proper metrics and user studies) and is not just overfitting will change my opinion on this paper siginificantly.

Additional questions:
- It is mentioned that the position set P models the buildings by their mean centers, however, judged by later sections, it seems that phase 1 is instead generating an occupancy map over locations, instead of a map over the building centers. Could the authors clarify what exactly happens in phase 1? If the model indeed predicts occupancy, then more analysis is needed on how it is converted into positions (as mentioned in Appendix D)
- The distribution predicted by phase 1 also seems extremely blurry and doesn't really resemble a probability distribution. It almost seems to me that the method is just attempting to memorize the silhouettes of buildings. A bit more analysis would be great here.
- In section 4, it mentioned that weights need to be applied to BCE to address class imbalance, this shouldn't happen if the model can actually learn the distribution. It seems that after applying such weights, there isn't really too much difference between high/low likelihood, which is not ideal.
- Finally, I am not sure if it make sense to learn the entire distribution over many buildings: shouldn't the distribution just be completely uniform since all buildings can take all locations without additional constraints? Shouldn't the probability be zero over locations with existing buildings? Some clarifications are again needed.

---

> ### Author Response · Authors · 2023-11-22
> **Add a new metirc CitySim, a novelness score, and a uniqueness score that verify our method is able to generate high quality layout while not overfitting. Part I**
>
> Thank you for the constructive feedback, and we provide some discussion and responses regarding your comments.
> 1. “Not too much novelty: the idea of predicting a probability distribution over building, the idea of autoregressive generation of polygons and the idea of transformer based layout generation can all be traced to prior works. While the domain (city generation) and the combination of techniques are novel (conditioning polygon generation directly on the location distribution), such novelty are likely not directly useful for people who are not interested in this particular problem. Subsequently, results quality would matter much more and. ”
>
> **Answer:** We acknowledge your concern that some of our techniques may be traced back to previous works. However, we would like to clarify the novelty of our approach.
>
> Firstly, in the task setting, we aim to generate city layouts with **arbitrary scales and buildings of arbitrary shapes**, a dimension that has not been thoroughly explored in city layout generation tasks or other more general layout generation tasks. We believe that our exploration in this area, specifically in arbitrary-shaped polygon layout generation, can inspire further research.
>
> Secondly, our approach possesses its own novelty. While there are existing works that utilize transformers for layout generation, to the best of our knowledge, all these transformer-based models **require predefining the order of the layout**. For example, some works define the order of each box (in our case, polygons) based on the x-coordinate of the center of the box, and in cases where x-coordinates are the same, the y-coordinate is used. In contrast, **our approach eliminates the need for this predefined order** through the introduction of the positional prediction stage. This disordered generation with positional guidance is important in various applications, such as infinite-scale layout generation, city complementation, and more. We believe that our exploration in more general layout generation tasks is valuable.
>
> 2. “The evaluation is underwhelming. Most baselines are for general layout problems that has much weaker constraints than the specific problem. The one baseline that addresses this problem (AETree) is a weak one the doesn't even compete with more general methods. Even with this set of baselines, I am not sure whether the proposed method really generates better layout, based on the qualitative results. The quantitative metrics are too generic (FID is too general for evaluating such layout visualizations, WD over edge/area/ratio doesn't really evaluate layout quality), the user/perceptual study is also not well conducted (ground truth shouldn't have minor noises, visualizations of layout should be better so humans can actually judge if the layouts are good i.e. not just with blue boxes/meshlab screenshots).”
>
> **Answer:** Firstly, we acknowledge your concern about the baseline choice. Originally, we intended to compare our model with Globalmapper[2]. Unfortunately, they released their code close to the submission deadline. Given the non-trivial nature of adapting their method to our task quickly, we couldn't deploy their model as a baseline during the rebuttal time. However, we are open to including a comparison between their model and ours in future work.
>
> Secondly, regarding the metric question, we believe FID and WD are valid metrics, as many related works in the same research field still use these metrics. To better evaluate our model's generation results, we introduced a new metric called CitySim, inspired by READ[1]. Reviewer can find the results in Table 2, and the specific design of these metrics is detailed in Appendix C. Our method still outperforms baselines on this metric.
>
> Thirdly, for addressing reviewer’s concern about the user study, we provide new user study results in Section 4.2, specifically in cases where noise is not added to the ground truth. Additionally, we offer visual comparisons between ground truth with noise and ground truth in Appendix H. The results show that our generation results are most similar to the ground-truth among the baseline.
>
> [1] Patil, Akshay Gadi, et al. "Read: Recursive autoencoders for document layout generation." CVPR 2020.
>
> [2] He, Liu, and Daniel Aliaga. "GlobalMapper: Arbitrary-Shaped Urban Layout Generation." ICCV 2023.
>
> 3. “Lack of evaluation over whether the method can generate novel and diverse layouts that are different from the training set. I am not convinced that the model is not overfitting to some training samples.”
>
> **Answer:** To address the reviewer's concern about whether our model is overfitting to the training samples, we've included a discussion and relative scores in Appendix D. This aims to clarify that our model is not overfitting to any training samples. Specifically, we add a novelness and uniqueness metrics that verifies our method is able to generate novel and unique layouts instead of overfitting to the training data.

---

> ### Author Response · Authors · 2023-11-22
> **Add a new metirc CitySim, a novelness score, and a uniqueness score that verify our method is able to generate high quality layout while not overfitting. Part II**
>
> 4. “The problem setting isn't particularly useful: without streets, roads, building types and other city elements, I can't see how this can be helpful in any real city planning / modeling tasks. I am also not sure whether representing buildings as 2D polygon contours adds much over just specifying the location and size of buildings: one would need to model the 3D building in some other ways anyways.”
>
> **Answer:** We believe reviewer’s comments can provide sensible guidance for our future directions. We have demonstrated that our model has the ability to generate layouts based on road networks. Therefore, it is possible for us to integrate our model with other road generation models and create more useful city planning settings. Now, we would like to share some of our thoughts.
>
> Firstly, we believe that within the layout generation task itself, it can still convey additional information beyond the building shape. The space between two buildings in the generated layout typically forms a road structure, which serves as useful guidance for city planning.
>
> Secondly, there are many works that focus solely on city layout generation, such as BlockPlanner [1] and GlobalMapper [2].
>
> Thirdly, we believe that 2D polygon contours are more useful than boxes. For instance, when generating the entire city and aiming to create a 3D wireframe for each building, the polygon footprints can serve as a valuable prior. For example, using PolyGen [3] to generate each building's wireframe, its autoregressive generation process can easily incorporate the polygon footprints as a prior condition for the final generated results. Since our layout generation has captured the relationship between different footprints, downstream PolyGen can also capture the relationship between buildings through this prior polygon footprint layout, while using boxes may not achieve this as effectively.
>
> [1] Xu, Linning, et al. "BlockPlanner: city block generation with vectorized graph representation." ICCV 2021.
>
> [2] He, Liu, and Daniel Aliaga. "GlobalMapper: Arbitrary-Shaped Urban Layout Generation." ICCV 2023.
>
> [3] Nash, Charlie, et al. "Polygen: An autoregressive generative model of 3d meshes." ICML 2020.
>
> 5. “It is mentioned that the position set P models the buildings by their mean centers, however, judged by later sections, it seems that phase 1 is instead generating an occupancy map over locations, instead of a map over the building centers. Could the authors clarify what exactly happens in phase 1? If the model indeed predicts occupancy, then more analysis is needed on how it is converted into positions (as mentioned in Appendix D)”
>
> **Answer:** We apologize for any confusion caused by some of our explanations. The position set P represents the mean centers of the buildings, and in Appendix D (after revision is Appendix G), we simply discretizes the coordinates of each vertex in the polygons within the layouts.
>
> 6. "The distribution predicted by phase 1 also seems extremely blurry and doesn't really resemble a probability distribution. It almost seems to me that the method is just attempting to memorize the silhouettes of buildings. A bit more analysis would be great here."
>
> **Answer:** We apologize for any confusion arising from our explanation of visualization. The visualized positional map is not a normalized distribution across the entire canvas. Instead, it is a 50*50 grid map, with each value falling within the range of [0, 1]. This value represents the probability of the existence of a building at a given position on the map. During our sampling process, we normalize this 50*50 map using softmax and subsequently sample positions from it.

---

> ### Author Response · Authors · 2023-11-22
> **Add a new metirc CitySim, a novelness score, and a uniqueness score that verify our method is able to generate high quality layout while not overfitting. Part III**
>
> 7. "In section 4, it mentioned that weights need to be applied to BCE to address class imbalance, this shouldn't happen if the model can actually learn the distribution. It seems that after applying such weights, there isn't really too much difference between high/low likelihood, which is not ideal."
>
> **Answer:** We apologize for any confusion in our Stage 1 description. In each grid on the map, we are addressing a binary classification problem. We have applied Binary Cross-Entropy (BCE) loss with a positive weight, considering the much smaller number of buildings in each layout block compared to 50x50. We believe this is a commonly used method in binary classification to mitigate false negative scores. Our results demonstrate that this learned distribution can yield reasonable generation outcomes.
>
> 8. "Finally, I am not sure if it make sense to learn the entire distribution over many buildings: shouldn't the distribution just be completely uniform since all buildings can take all locations without additional constraints? Shouldn't the probability be zero over locations with existing buildings? Some clarifications are again needed."
>
> **Answer:** It is correct that there are no buildings in the canvas, so we generate the first position from a uniform distribution during our inference time. However, when there are existing buildings, the position of other buildings may be constrained by their presence. For instance, the existence of a building may suggest the occurrence of a road, as the orientation of the road often aligns with the orientation of the building. In such cases, the possibility of other buildings appearing in the implied road structure decreases. Given that our training procedure, iterative inference process, and many applications of city generation tasks involve generating buildings based on the condition of existing ones, the positional map becomes meaningful.
>
> It is correct that the probabilities are zero over locations with existing buildings. Our first-stage results have demonstrated a tendency to predict zero probabilities over locations with existing buildings. During our inference time, we validate the positional map by masking the locations with existing buildings, as discussed in our Section 3.4.

---

> > ### Comment · Reviewer_2WzD · 2023-11-22
> >
> > Thanks for the response to my concerns.
> >
> > 1. I get that the setting is new, as will be the case for most system papers. The point is whether any specific ideas introduced here can be useful for *other* tasks that's not city layout generation. If there are good ideas that can help other tasks (which I don't think so), then the result quality can be of a lesser concern - since this paper can be helpful even if the results are not directly applicable in production settings yet. On the other hand, if most of the novelty is limited to this particular setting (and I acknowledged this already), then more emphasis will be placed on results quality, and I don't think it is good.
> >
> > I disagree with the claim that existing transformers require predefining the order of the layout. A quick literature review should allow you to realize that many transformer-based generative models are permutation invariant - not only for layout generation, but also for other tasks such as image/shape generation. This claim, to be, clearly highlighted the authors lack of knowledge on related works.....
> >
> > 2.The user study w/o noise makes sense, a larger sample size than 21 will be helpful. Instead of presenting overall preference, pairwise preference (ours vs gt, layouttrans vs gt, ours vs layoutrans) will be more informative. Still, a better visualization style is needed to allow users to better compare layouts.
> >
> > I get the argument about FID/WD being acceptable metrics. My point is that good performance on them is necessary but not sufficient. To truly demonstrate that the results are good, more qualitative samples and user study are still needed (with the new w/o noise study, it seems that quality is still worse than gt but a good margin, quantifying how large of margin there is would be helpful). The new metric is nice, but still does not show that the proposed method have a clear edge over non-domain-specific approaches i.e. LayoutTrans.
> >
> > 3. Appendix D is nice, but average distance to train/val type of metrics only show a rough picture, it's hard to decipher "how much overfitting there is" by simply looking at these numbers. Qualitative examples showing that a certain layout never appears in the training set is still necessary. Also, since a lot of distributions are predicted, it will be easier to gauge the amount of overfitting by visualizing the distributions.
> >
> > 4. Re: 2D polygon contours: an ablation showing that it indeed helps over just boxes would be great here.
> >
> > 5. I get that it's mean centers but shouldn't distrubution for centers form a much thinner line (as opposed to being as thick as the entire building)?
> >
> > 6. "probability of the existence of a building": i am confused, is it existence of building or "center of building"?
> >
> > 7. What happens without weights? Why will there be "false negatives" if you are just sampling centers from a probability distribution? Even if the maximum probability anywhere is 10% (which should be the case considering that centers are sparse), you can still reliably sample a center from this distribution. This is only an issue if you are thresholding an occupancy map, but this seems to not be the case based on the descrption of phase 1?
> >
> > 8. It would be nice to show how this probability map evolves from 0 building, to a few buildings, and to a lot of existing buildings. The current stage 1 visualizations seems to be about probability distributions when there are lots of existing buildings. This distribution should look very different when, say, only a single building exist in the input. This visualization is *necessary* for me to be convinced that phase 1 is actually learning a distribution, as opposed to overfitting to shapes.
> >
> > "Our first-stage results have demonstrated a tendency to predict zero probabilities over locations with existing buildings.": how so? I don't see it in the examples given.

---

> ### Author Response · Authors · 2023-11-23
> **Our model is not limited to the city layout generation task.**
>
> 1. Firstly, we need to mention that our method is not limited to city layout generation. It is clear that our model can be applied to various layout generation tasks, such as document layout and more. Our model is a more general case compared to existing layout generation tasks, as it can generate arbitrary scale layouts with arbitrary shapes. As mentioned in the comments of reviewers YWWm (“This paper is the first to represent layouts of arbitrary scales and shapes without any prior conditions. The proposed two-stage decomposition modeling approach for city layout can accomplish various layout generation tasks.”) and reviewer K6Hc (“The two-stage approach with MAE-like training is novel and breaks down a hard problem into iterative generation of layout and geometry. The autoregressive transformer is also more capable as it can generate arbitrary polygon shapes as opposed to just the 2D bounding boxes.”).
>
> Secondly, we believe there might be some misunderstanding of relevant methods regarding permutation invariance. We want to point out that many transformer-based layout generation models (including LayoutTrans) autoregressively generate layouts, requiring a predefined order (as we claimed in our paper). We are not the first to point out the immutable problem in LayoutTrans and some other transformer-based layout generation models. Many diffusion-based works have also pointed out this problem, as seen in the Introduction part of LayoutDM [1] and the related work part of LayoutDiffusion [2].
> Because the self-attention in transformers has the permutation invariance property, existing generative transformers require positional embeddings to break that permutation invariance so as to generate ordered data such as texts and images. This makes it non-trivial to develop a generative transformer for unordered data (such as building/document layouts). Existing methods typically choose from two ways.One way is to use the autoregressive transformer by assuming some (often manually defined) orders or hierarchies in the data which is not necessarily a good assumption. Many works in layout generation methods are like this, bringing about the problem we mentioned above.
> Another way is to  generate the whole data in one shot, and then perform matching between generation results and the ground truth data when computing losses. For example, DETR [3] uses hungarian matching to map the generated results to the ground truth, but  this method is proved to be hard to converge by following works, and our Appendix F also demonstrates that this matching method cannot accomplish our task. Some works tend to solve this problem by designing heuristic rules like DeepSVG [4] and BLT [5].
> In fact, this challenging and non-trivial problem is addressed by our method, and it can easily extend to various application tasks (conditional generation tasks) in layout generation (not limited to city layout).
>
> [1] Inoue, Naoto, et al. "LayoutDM: Discrete Diffusion Model for Controllable Layout Generation." Proceedings of the IEEE/CVF Conference on Computer Vision and Pattern Recognition. 2023.
>
> [2] Zhang J, Guo J, Sun S, et al. LayoutDiffusion: Improving Graphic Layout Generation by Discrete Diffusion Probabilistic Models[J]. arXiv preprint arXiv:2303.11589, 2023.
>
> [3] Carion, Nicolas, et al. "End-to-end object detection with transformers." European conference on computer vision. Cham: Springer International Publishing, 2020.
>
> [4] Carlier, Alexandre, et al. "Deepsvg: A hierarchical generative network for vector graphics animation." Advances in Neural Information Processing Systems 33 (2020): 16351-16361.
>
> [5] Kong, Xiang, et al. "BLT: bidirectional layout transformer for controllable layout generation." European Conference on Computer Vision. Cham: Springer Nature Switzerland, 2022.

---

> ### Author Response · Authors · 2023-11-23
>
> 2. We acknowledge that pairwise comparison can be a good choice, but we think our setting, choosing the best from three, also makes enough sense. We believe that our visualization of 2D layout is good enough; it visualizes all the information contained in the layout. If there are some better ideas, thanks for pointing them out.
>
> 3. We believe the histogram effectively conveys the extent to which our model overfits. It illustrates that the maximum PixelSim score over all 1000*32000 pairs is less than 95%. Additionally, for the majority of the 1000 generated samples, the nearest layout in the training data only has a 60% similarity. So, it is clear that our generated results are not overfit to the training set.
>
> 4. We agree with your idea that experiments can show things more confidently. However, we believe it is clear that polygons can convey more information compared to boxes, which is useful in downstream tasks. As mentioned in the previous reply, if we concatenate the layout generation with a single 3D object generation model, polygons bring much more prior information.
>
> 5. We need to point out that since city layouts are not often a straight line with the same shape, some buildings may have different shapes (longer than others on some sides), causing their centers to vary widely, especially in a polygonal setting. Thus, a building can appear in a relatively large area with similar probability. We acknowledge your concern that, in the most optimistic case, the distribution can be predicted in some certain points with very high probability, allowing for the generation of the entire city in one shot. However, we find it difficult to accomplish this due to the reasons mentioned above (variation in building shapes and centers). Therefore, in our inference stage, we apply an iterative generation pipeline.
>
> 6. A more explicit expression is "the probability of the existence of a building centered at that grid". Note that we do not constrain the generated polygon to exactly center in this grid; we find that our model can capture this information, and the generated polygon centers are always located near this corresponding grid.
>
> 7. Without the weight, it is true that our model can still generate layouts but not as effectively as in the current setting. In the training stage, without applying this positive weight, the positions where existing buildings are present cannot be well backpropagated (due to their small proportion in the gradient). This makes our model struggle to understand the existence of buildings or, if it can, it still has difficulty converging.
>
> 8. We show these step by step results in Appendix H. "Our first-stage results have demonstrated a tendency to predict zero probabilities over locations with existing buildings." For more details, please refer to the fourth row in Figure 4, where you can observe that the positions with purple buildings are predicted as black (indicating a low probability of having buildings).

---

### Author Response · Authors · 2023-11-22
**To all reviewrs: thank you all for the constructive feedbacks and encouragements.**

We sincerely appreciate all reviewers' constructive feedback and positive comments: (a) The paper addresses an interesting problem of generating layouts of arbitrary scales and shapes without any prior conditions and provides an approach that can enable a range of applications. (Reviewer 2WzD and YWWm) (b) Our experimental results prove the effectiveness of the proposed method, performing better than some of the baselines. (Reviewer 2WzD, YWWm, and zw2r) (c) Our two-stage approach with MAE-like training is novel, and experiments show the necessity of the two-stage design. (Reviewer 2WzD, YWWm, K6Hc)

Based on the reviewers' comments, we have revised the manuscript, and the changes are highlighted in blue text. The major modifications we have made include:
1. Add additional metrics (**CitySim**) that shows our method is able to generate higher quality of layout than baselines.
2. Add additional novelness and uniqueness metrics that verifies our method is able to generate novel and unique layouts instead of overfitting to the training data.
3. Conduct a new user study where the minor noises are removed. The results show that our generation results are most similar to the ground-truth.
4. Provide more detailed explanations of our inference pipeline.
5. Provide more detailed explanations of the implementation of further experiments.
6. Add a runtime performance analysis of our inference stage, which shows reasonable inference efficiency of our method compared to baselines.
7. Add some discussion about more related works and address some typos.

We have addressed all reviewers’ questions and improved our paper according to reviewers’ suggestions on the evaluation and experiments. The overall results of experiments are still valid when introducing the new evaluation metrics. And our method could not only generate higher quality layouts than baselines, it could  also generate novel layouts instead of overfitting to the training data. Additionally, we have provided detailed responses to each reviewer's comments individually in the following sections.

---

### Meta-Review · Area_Chair_ZPSw · 2023-12-11

**Metareview:**

This paper introduces CityGPT, a generative pretrained transformer for city layout generation. It alternates between two steps, with one learning to first  predict building center positions conditioned on unmasked layouts and the other learning to estimate masked layouts conditioned on their sampled center positions and unmasked layouts. It also incorporates an autoregressive polygon model to handle arbitrary building shapes. The experimental results demonstrate the effectiveness of the proposed method to some extent on several generation tasks.

Strengths:
The proposed two-stage pipeline that breaks down a hard problem into iterative generation of layout and geometry is novel and reasonable.

Weaknesses:
Novelty of this work for layout generation is limited except for the two-stage pipeline.
The experimental results are weak and not very convincing.

**Justification For Why Not Higher Score:**

Several reviewers have concern about the novelty and performance evaluation.

**Justification For Why Not Lower Score:**

This work is recommended to be rejected.

---

### Decision · Program_Chairs · 2024-01-16

Reject